# Observation of the scaling dimension of fractional quantum Hall anyons

A. Veillon[1], C. Piquard[1], P. Glidic[1], Y. Sato[1], A. Aassime[1], A. Cavanna[1], Y. Jin[1], U. Gennser[1], A. Anthore[1,2 ✉] & F. Pierre[1 ✉]

Unconventional quasiparticles emerging in the fractional quantum Hall regime[1,2] present the challenge of observing their exotic properties unambiguously. Although the fractional charge of quasiparticles has been demonstrated for nearly three decades[3–5], the first convincing evidence of their anyonic quantum statistics has only recently been obtained[6,7] and, so far, the so-called scaling dimension that determines the propagation dynamics of the quasiparticles remains elusive. In particular, although the nonlinearity of the tunnelling quasiparticle current should reveal their scaling dimension, the measurements fail to match theory, arguably because this observable is not robust to non-universal complications[8–12]. Here we expose the scaling dimension from the thermal noise to shot noise crossover and observe an agreement with expectations. Measurements are fitted to the predicted finite-temperature expression involving both the scaling dimension of the quasiparticles and their charge[12,13], in contrast to previous charge investigations focusing on the high-bias shot-noise regime[14]. A systematic analysis, repeated on several constrictions and experimental conditions, consistently matches the theoretical scaling dimensions for the fractional quasiparticles emerging at filling factors $v = 1/3$, 2/5 and 2/3. This establishes a central property of fractional quantum Hall anyons and demonstrates a powerful and complementary window into exotic quasiparticles.

Exotic quasiparticles could provide a path to protected manipulations of quantum information[15]. Yet their basic features are often challenging to ascertain experimentally. The broad variety of quasiparticles emerging in the regimes of the fractional quantum Hall effect constitutes a prominent illustration. These are characterized by three unconventional properties[1,2]: (1) their charge $e^*$ is a fraction of the elementary electron charge $e$; (2) their anyonic quantum statistics is different from that of bosons and fermions; and (3) the dynamical response to their injection or removal along the propagative edge channels is peculiar, ruled by a 'scaling dimension' $\Delta$ different from the trivial $\Delta = 1/2$ of non-interacting electrons. In the simplest Laughlin quantum Hall states, at filling factors $v = 1/(2n + 1)$ ($n \in \mathbb{N}$), the fractional anyon quasiparticles have a charge $e^* = ve$, an exchange phase $\theta = v\pi$ and a scaling dimension $\Delta = v/2$. Despite four decades of uninterrupted investigations of the quantum Hall physics, experimental confirmations of the predicted scaling dimension remain lacking, including for Laughlin fractions.

Such a gap may seem surprising because $\Delta$ plays a role in most transport phenomena across quantum point contacts (QPCs), the basic building block of quantum Hall circuits. Indeed, the elementary tunnelling process itself consists in the removal of a quasiparticle on one side of a QPC and its reinjection on the other side, whose time correlations are set by $\Delta$ (refs. 1,2). In Luttinger liquids, the scaling dimension of the quasiparticles is related to the interaction strength, also referred to as the interaction parameter $K$, which notably determines the nonlinear

$I–V$ characteristics across a local impurity[16]. Consequently, the knowledge of $\Delta$ is often a prerequisite to connect a transport observable with a property of interest. Furthermore, as straightforwardly illustrated in the Hong–Ou–Mandel set-up[17–19], $\Delta$ naturally rules time-controlled manipulations of anyons, which are required in the perspective of topologically protected quantum computation based on braiding[15]. In this work, the scaling dimension of fractional quantum Hall quasiparticles is disclosed from the thermal noise to shot noise crossover, as recently proposed[12,13]. The observed good agreement with universal predictions establishes experimentally the theoretical understanding and completes our picture of the exotic fractional quantum Hall anyons.

## Characterizing exotic quasiparticles

The first unconventional property of quantum Hall quasiparticles that has been established is their fractional charge $e^*$. Consistent values were observed by several experimental approaches[3–5,20–26], with the main body of investigations based on shot-noise measurements across a QPC. In this case, the scaling dimension can be cancelled out, leaving only $e^*$, by focusing on the ratio between shot noise and tunnelling current (the Fano factor) at high bias voltages. The non-standard braiding statistics of fractional quasiparticles turned out to be more challenging to observe. Convincing evidences were obtained only recently, through Aharonov–Bohm interferometry[6,27], as well as from

[1]Université Paris-Saclay, CNRS, Centre de Nanosciences et de Nanotechnologies, Palaiseau, France. [2]Université Paris Cité, CNRS, Centre de Nanosciences et de Nanotechnologies, Palaiseau, France. ✉e-mail: anne.anthore@c2n.upsaclay.fr; frederic.pierre@cnrs.fr

noise measurements in a 'collider' geometry[7,28–30]. Note that, whereas the latter strategy is particularly versatile, the noise signal is also entangled with the scaling dimension[18,19,31,32], which complicates a quantitative determination of the anyon exchange phase $\theta$ (ref. 28). Finally, the scaling dimension of the quasiparticles was previously investigated through measurements of the nonlinear $I$–$V$ characteristics of a QPC[33–35]. However, no reliable value of $\Delta$ could be obtained for the fractional quasiparticles of the quantum Hall regime (see ref. 36 for an observation in a circuit quantum simulator and ref. 37 for a good match on the $I(V)$ of tunnelling electrons across a $(\nu = 1)$–$(\nu = 1/3)$ interface). Indeed, the $I$–$V$ characteristics is generally found at odds with the standard model of a chiral Luttinger liquid with a local impurity (see, for example, refs. 2,14,33,38 and references therein). Most often, a fit is impossible or only by introducing extra offsets and with unrealistic values for $e^*$ and $\Delta$ (refs. 29,34,35,39).

The puzzling $I$–$V$ situation motivated several theoretical investigations. A simple possible explanation for the data–theory mismatch is that the shape of the QPC potential, and therefore the quasiparticle tunnelling amplitude, is affected by external parameters, such as an electrostatic deformation induced by a change in the applied bias voltage, the temperature or the tunnelling current itself[10]. Other possible non-universal complications include an energy-dependent tunnelling amplitude[11], further edge modes either localized[8] or propagative[40,41] and Coulomb interactions between different edges[9]. In this context, the scaling dimension was connected to different, arguably more robust proposed observables, such as delta-$T$ noise[42,43], thermal noise to shot noise crossover[12,13] and thermal Fano factor[44].

A proven strategy to cancel out non-universal behaviours consists in considering a well-chosen ratio of transport properties, as illustrated by the Fano factor $F$ successfully used to extract $e^*$. Recently, it was proposed that the same $F$ could also give access to the scaling dimension of the quasiparticles, when focusing on the lower bias voltage regime in which the crossover between thermal noise and shot noise takes place[12,13]. As further detailed later on, the predicted evolution of $F$ along the crossover exhibits a markedly different width and overall shape depending on the value of $\Delta$.

This investigation implements the characterization of the scaling dimension from the Fano factor crossover on four different quantum Hall quasiparticles: (1) the $e^* = e/3$ quasiparticles, observed at $\nu = 1/3$ (ref. 4) and along the outer edge channel of conductance $e^2/3h$ at $\nu = 2/5$ (refs. 20,24), of predicted $\Delta = 1/6$ (ref. 1); (2) the $e^* = e/5$ quasiparticles observed along the inner edge channel of conductance $e^2/15h$ at $\nu = 2/5$ (refs. 20,24), of predicted $\Delta = 3/10$ (refs. 1,45); (3) the $e^* = e/3$ quasiparticles observed at $\nu = 2/3$ (refs. 25,46), of predicted $\Delta = 1/3$ (ref. 45); (4) the electrons at $\nu = 3$ of trivial $\Delta = 1/2$. See Methods for further details on the predictions.

## Experimental implementation

The measured sample is shown in Fig. 1, together with a schematic representation of the set-up. It is nanofabricated from a Ga(Al)As two-dimensional electron gas (2DEG) and immersed in a strong, perpendicular magnetic field corresponding to the quantum Hall effect at filling factors $\nu \in \{1/3, 2/5, 2/3, 3\}$. Lines with arrows show the chiral propagation of the current along the sample edges. QPCs are formed in the 2DEG by field effect, within the opening of metallic split gates (yellow). We characterize a QPC by the gate-controlled transmission ratio $\tau \equiv I_B/I_{inj}$, with $I_B$ the backscattered current and $I_{inj}$ the incident current along the edge channel under consideration. The sample includes five QPCs nominally identical except for their orientation and the presence for QPC$_W$ of a surrounding gate (labelled TG in the inset of Fig. 1). This surrounding gate allows us to test the possible influence on the scaling dimension of the local 2DEG density, of an enhanced screening of the long-range Coulomb interactions[9] and of an increased sharpness of the electrostatic edge-confinement potential[40].

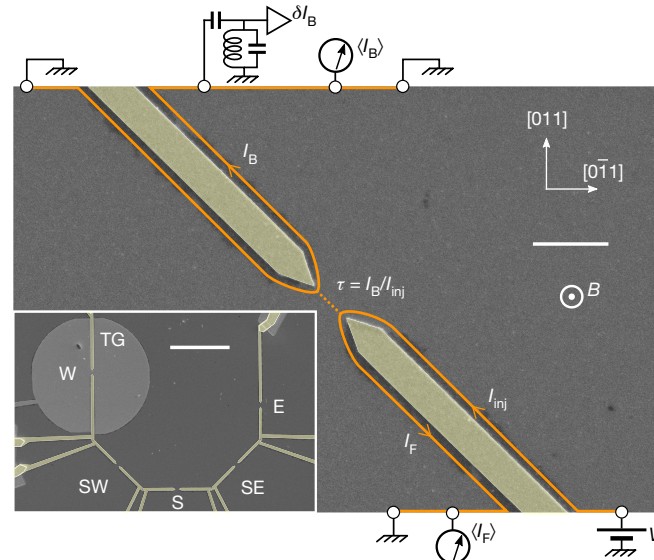

**Fig. 1 | Experimental set-up.** Electron-beam micrographs of the measured Ga(Al)As device. QPCs are formed in the 2DEG by applying a negative voltage to the metallic gates coloured yellow. The sample includes five QPCs (E, SE, S, SW and W) along different crystallographic orientations (inset; see Extended Data Fig. 1 for larger-scale images; scale bar, 10 μm). Among those, QPC$_W$ differs by the presence of a closely surrounding metallic gate (TG) extending over a 10-μm radius (see Extended Data Fig. 2 for close-up images). Quasiparticle tunnelling takes place between chiral quantum Hall edge channels shown as orange lines with arrows (main panel, QPC$_{SW}$; scale bar 1 μm). The autocorrelations in backscattered (tunnelling) current $\langle \delta I_B^2 \rangle$ are measured for all QPCs. QPC$_E$ also includes a noise-amplification chain for the forward current fluctuations $\delta I_F$ (not shown), hence allowing for the extra measurements of $\langle \delta I_F^2 \rangle$ and $\langle \delta I_B \delta I_F \rangle$.

The noise is measured using two cryogenic amplifiers (one is shown schematically). The gain of the noise-amplification chains, and the electronic temperature within the device, were obtained from their relation to thermal noise (Methods). One amplifier (top left in Fig. 1) measures the backscattered (tunnelling) current noise $\langle \delta I_B^2 \rangle$ for any addressed QPCs. A second amplifier (not shown) measures the forward current fluctuations $\delta I_F$ transmitted specifically across QPC$_E$. In practice, we focus on the excess noise with respect to zero bias: $S(V) \equiv \langle \delta I^2 \rangle(V) - \langle \delta I^2 \rangle(0)$.

## Scaling-dimension characterization

In previous characterizations of the charge $e^*$ of fractional quantum Hall quasiparticles, the shot noise is usually plotted as a function of the backscattered current $I_B$ and $e^*$ is extracted by matching the high-bias slope $\partial S/\partial I_B$ with $2e^*(1 - \tau)$, in which $1 - \tau$ corrects for tunnelling correlations at finite $\tau$ (ref. 47). Even in this representation, which puts the emphasis on the larger high-bias shot noise, a visually discernible and experimentally relevant difference allows us to discriminate between predicted and trivial $\Delta$, as illustrated in Fig. 2a–c. Continuous blue lines show the excess shot noise of quasiparticles of trivial $\Delta = 1/2$ and of charge $e/3$ (Fig. 2a,c) or $e/5$ (Fig. 2b), which is given by the broadly used phenomenological expression[14,47]:

$$S_{1/2} = 2e^* I_B (1 - \tau) \left[ \coth \frac{e^* V}{2k_B T} - \frac{2k_B T}{e^* V} \right]. \quad (1)$$

Note that, for composite edges with several electrical channels (such as $\nu \in \{2/5, 3\}$), $\tau \equiv I_B/I_{inj}$, in which $I_B$ and $I_{inj}$ refer to the dc transmission ratio and currents along the specific edge channel of interest (Methods). The continuous lines of a different colour in the main panels of

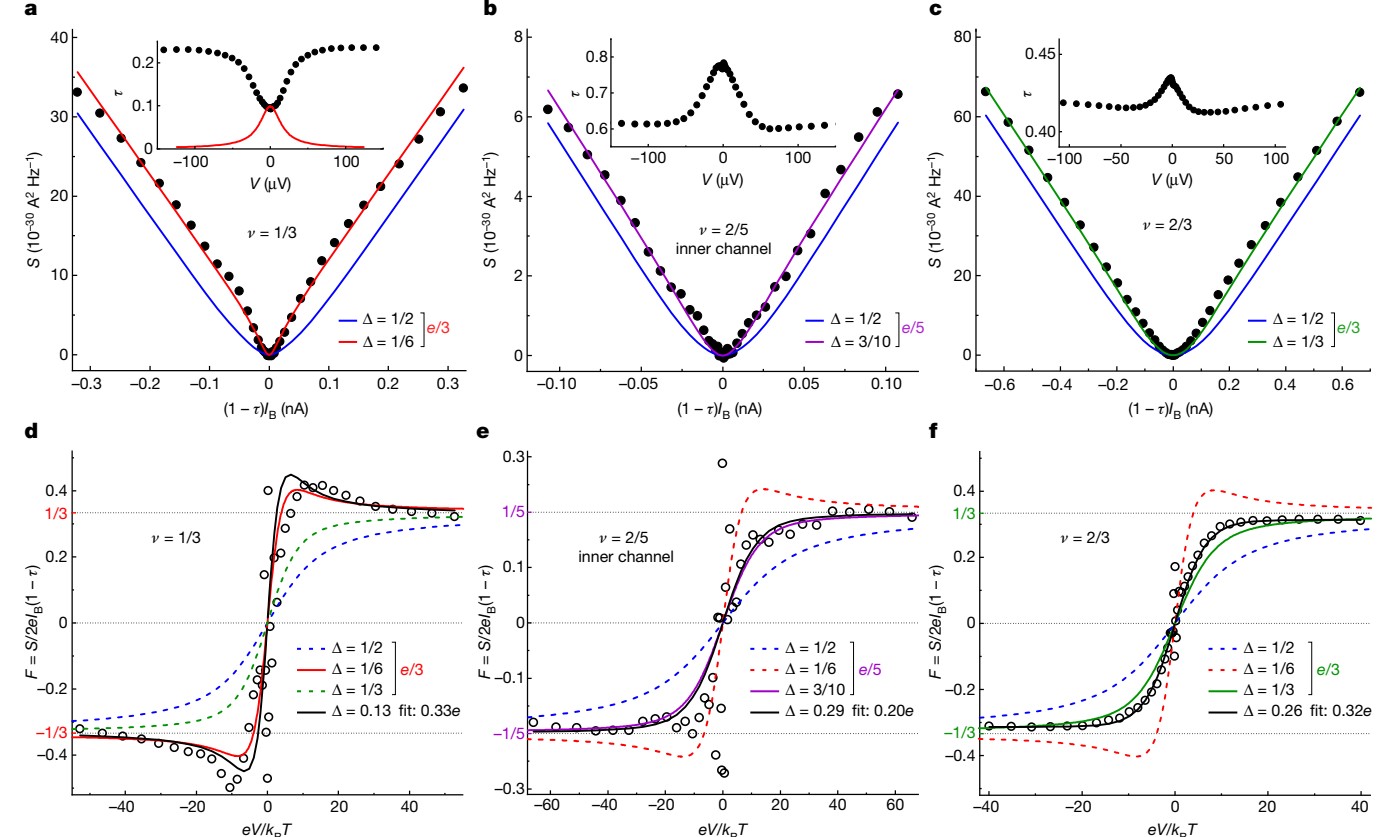

**Fig. 2 | Thermal noise to shot noise crossover.** Data–theory comparison at $\nu = 1/3$ (**a**,**d**), on the inner $e^2/15h$ channel of $\nu = 2/5$ (**b**,**e**) and at $\nu = 2/3$ (**c**,**f**). **a**–**c**, Excess noise $S$ versus normalized tunnel current $(1 - \tau)I_B$. Symbols are measurements (**a**,**d**, QPC$_W$ with $V_{tg} = 50$ mV at $T \simeq 30.85$ mK; **b**,**c**,**e**,**f**, QPC$_E$ at $T \simeq 30.7$ mK), with a standard error on the noise of $1 \times 10^{-31}$ A$^2$ Hz$^{-1}$, which is smaller than the size of the symbols. Blue lines are phenomenological predictions of equation (1) ($\Delta = 1/2$, predicted $e^*$). Red, purple and green lines

are predictions of equation (2) (predicted $\Delta$ and $e^*$). Insets, $\tau(V)$ measurements are shown as symbols. The $\nu = 1/3$ prediction (red line) differs strongly from these. **d**–**f**, Fano factor $F \equiv S/2eI_B(1 - \tau)$ versus $eV/k_B T$. Measurements (symbols) agree best with the predictions of equation (2) computed using the predicted quasiparticle scaling dimension $\Delta$ (coloured continuous lines) than using the electron scaling dimension $1/2$ (dashed blue lines), both assuming the predicted $e^*$. Continuous black lines are fits using $e^*$ and $\Delta$ as free parameters.

Fig. 2a–c show the excess noise for the predicted quasiparticle scaling dimension $\Delta = 1/6$ (red, Fig. 2a), $\Delta = 3/10$ (purple, Fig. 2b) and $\Delta = 1/3$ (green, Fig. 2c) obtained from[8,12,13]:

$$S_\Delta = 2e^* I_B (1 - \tau) \mathrm{Im}\left[ \frac{2}{\pi} \psi\left( 2\Delta + i\frac{e^* V}{2\pi k_B T} \right) \right]. \tag{2}$$

Here $\psi$ is the digamma function and $1 - \tau$ the ad hoc amplitude factor used for extracting $e^*$ from the shot-noise slope at high bias (beyond the perturbative limit $\tau \ll 1$ in which equation (2) is rigorously derived). Whereas equation (2) reduces to equation (1) for $\Delta = 1/2$, for smaller $\Delta$, the shot noise emerges at a lower voltage. Intuitively, this can be connected through the time–energy relation to the slower decay of correlations at long times (as $t^{-2\Delta}$). For the quasiparticles $\{e/3, \Delta = 1/6\}$ predicted at $\nu = 1/3$, the apparent width of the crossover is more than twice narrower than for $\Delta = 1/2$ (Fig. 2a). The difference is smaller for the quasiparticles $\{e/5, \Delta = 3/10\}$ and $\{e/3, \Delta = 1/3\}$ because $\Delta$ is closer to $1/2$ (Fig. 2b,c). Nevertheless, as can be straightforwardly inferred from the scatter of the data, it remains in all cases larger than our experimental resolution on the noise. We can already notice that the illustrative shot-noise measurements shown in Fig. 2a–c are closer to the parameter-free prediction of equation (2) with the expected $\Delta$. Note that this agreement is accompanied by a puzzling $I$–$V$ characteristics as previously mentioned (see $\tau(V)$ in insets and also in Extended Data Fig. 9).

For the present aim of characterizing $\Delta$ from the thermal noise to shot noise crossover, the Fano factor $F \equiv S/2eI_B(1 - \tau)$ of bounded amplitude

at high bias is better suited[12,13]. It is plotted against the relevant variable $eV/k_B T$ (see equation (2)) in Fig. 2d–f, with symbols and coloured lines corresponding to the noise shown in the panel immediately above. Notably, the effect of $\Delta < 1/2$ on $F$ is not limited to an increased slope at low bias, which could—in principle—be attributed to a temperature lower than the separately characterized $T$, but results in marked changes in the overall shape of $F(eV/k_B T)$. In particular, for $\Delta = 1/6$, the Fano factor is non-monotonous (red line in Fig. 2d). The increasing steepness while reducing $\Delta$ combined with an overall change of shape enables the extraction of this parameter from a fit using equation (2). Qualitatively, the value of $F$ at large bias only reflects $e^*/e$, the overall crossover shape (such as a non-monotonous dependence at $\Delta < 1/4$) only involves $\Delta$ and the low-bias slope is a combination of both $e^*$ and $\Delta$. The results of such fits (minimizing the data equation (2) variance) are shown as black continuous lines in Fig. 2d–f, together with the corresponding fitting parameters $e^*$ and $\Delta$ (the temperature being fixed to the separately determined $T \simeq 31$ mK).

## Robustness of observations

Focusing on the Fano factor cancels out some of the non-universal behaviours, but not all of them. Of particular concern are the disorder-induced resonances, which could result in a Coulomb-dominated sequential tunnelling with a strong effect on the Fano factor. This probably happens in the fractional quantum Hall regime in which QPCs often exhibit narrow peaks and dips in their transmission $\tau$ versus gate

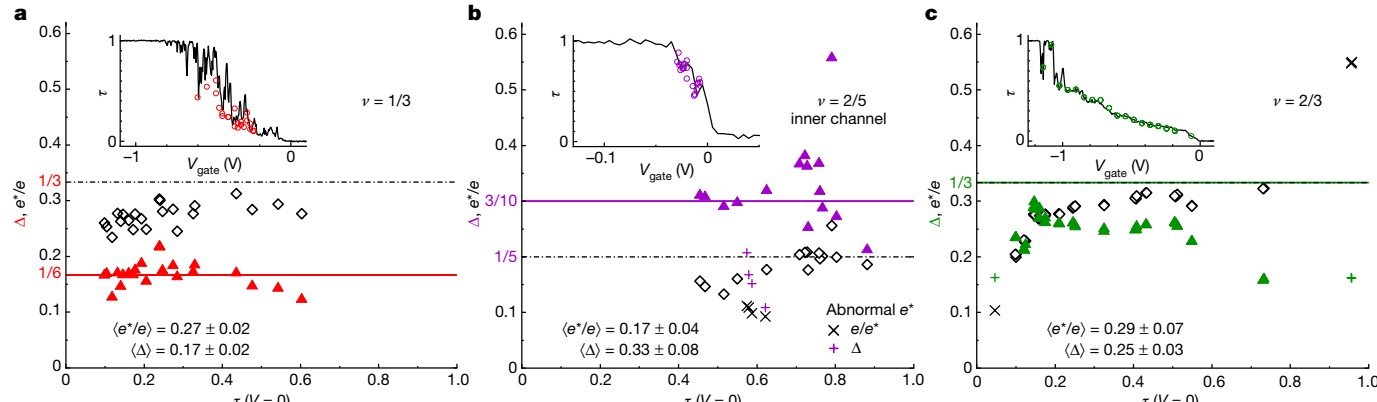

**Fig. 3 | Scaling dimension versus QPC tuning.** Scaling dimension (triangles) and charge (diamonds) obtained along illustrative spans in gate voltage ($V_{gate}$) of QPC$_E$ are plotted versus $\tau(V=0)$ (**a**, $\nu=1/3$; **b**, inner channel at $\nu=2/5$; **c**, $\nu=2/3$). At $\nu=2/5$ (**b**) and at $\nu=2/3$ (**c**), a few points are shown as different symbols (+, ×). They are associated with anomalously low $e^* \lesssim e_{th}^*/2$ or high $e^* \gtrsim 3/2e_{th}^*$ charge compared with the predicted charge $e_{th}^*$. Horizontal lines represent the theoretical predictions for $\Delta$ (solid lines) and $e^*/e$ (dashed-dotted lines). Insets, separately measured $\tau(V_{gate})$ sweeps (continuous lines) and individual noise-measurement tunings (symbols). For $\nu=1/3$, a noticeable difference results from a slightly different magnetic field setting ($\delta B \simeq -0.5$ T) between $V_{gate}$ sweep and noise measurements.

voltage (see insets in Fig. 3). Accordingly, for some gate voltages, we find that an accurate fit of the noise data is not possible with equation (2), whatever $e^*$ and $\Delta$. In such cases, the resulting fitted values are meaningless. This was transparently addressed with a maximum variance criteria between data and best fit. If the fit-data variance is higher, the extracted $e^*$ and $\Delta$ are discarded (see Methods). This same procedure was systematically applied to all the noise measurements performed over a broad span of gate voltages controlling $\tau$ (the full dataset, including discarded fits and analysis code, is available in a Zenodo deposit).

The values of $e^*$ and $\Delta$ obtained while spanning the gate voltage of the same QPC$_E$ are shown versus $\tau(V=0)$ in Fig. 3 for each of the three investigated fractional quasiparticles (see Methods for electrons at $\nu=3$). We find remarkably robust scaling dimensions (and charges) close to the predictions, shown as horizontal lines. In particular, although the nature of the tunnelling quasiparticles is eventually going to change at $\tau \to 1$, we observe that $\Delta$ and $e^*$ extracted with equation (2) (which is exact only at $\tau \ll 1$) remain relatively stable over a broad range of $\tau$. Such a stability matches previous $e^*$ measurements, including a particularly steady $e/5$ (ref. 48). Figure 3a shows data points obtained in the $\nu=1/3$ plateau. A statistical analysis of the ensemble of these points yields $\langle\Delta\rangle \simeq 0.167$, with a spread of $\sigma_\Delta \simeq 0.023$, which is to be compared with the prediction $\Delta=1/6 \simeq 0.1667$. The data-prediction agreement on $\Delta$ is at the level of, if not better than, that on $e^*$ (often found slightly lower than expected). Similar sweeps are shown in Fig. 3b,c for the inner channel of conductance $e^2/15h$ at $\nu=2/5$ (Fig. 3b) and at $\nu=2/3$ (Fig. 3c). Note that a few data points at $\nu=2/5$ and at $\nu=2/3$ are shown as pairs of '×' ($e^*/e$) and '+' ($\Delta$) instead of open and closed symbols (Fig. 3b,c). This indicates an anomalous fitted value of the charge $e^*$, off by about 50% or more from the well-established prediction $e_{th}^* = e/5$ and $e_{th}^* = e/3$, respectively (dashed-dotted lines). Because this suggests a non-ideal QPC behaviour (for example, involving localized electronic levels), we chose not to include these relatively rare points in the data ensemble analysis of $\Delta$ (they remain included in the statistical analysis of $e^*$). For this reduced dataset composed of 15 measurements along the inner channel at $\nu=2/5$, we obtain $\langle\Delta\rangle \simeq 0.327$ ($\sigma_\Delta \simeq 0.078$), which is to be compared with the predicted $\Delta=3/10$ of $e/5$ quasiparticles. Last, at $\nu=2/3$, the gate-voltage sweep shown in Fig. 3c gives $\langle\Delta\rangle \simeq 0.249$ ($\sigma_\Delta \simeq 0.029$), close to the predicted $\Delta=1/3 \simeq 0.33$. Note, however, that in this more complex case, with counterpropagating edge modes and the emergence of a small plateau versus gate voltage at $\tau \simeq 0.5$ (inset in Fig. 3c), the noise interpretation is not as

straightforward, especially when $\tau$ is not small (see Methods for further tests and discussions).

The robustness and generic character of these $\Delta$ observations are further established by repeating the same procedure in different configurations: (1) on several QPCs, with different orientations with respect to the Ga(Al)As crystal; (2) for several temperatures $T$; (3) for several top-gate voltages $V_{tg}$ controlling the density in the vicinity of QPC$_W$; (4) by changing the magnetic field, both along the $\nu=1/3$ plateau and to $\nu=2/5$ on the outer edge channel. Figure 4 recapitulates all our measurements (283 in total), including conventional electrons at $\nu=3$. Each point represents the average value $\langle e^*/e\rangle$ (diamonds) or $\langle\Delta\rangle$ (triangles) and the corresponding standard deviation obtained while broadly spanning the gate voltage of the indicated QPC (individually extracted $e^*$ and $\Delta$ are provided in Methods). See also Methods for consistent conclusions from an alternative fitting procedure in which $\Delta$ is the only free parameter ($e^*$ being fixed to the well-established prediction and focusing on low voltages $e^*|V| \le 2k_BT$).

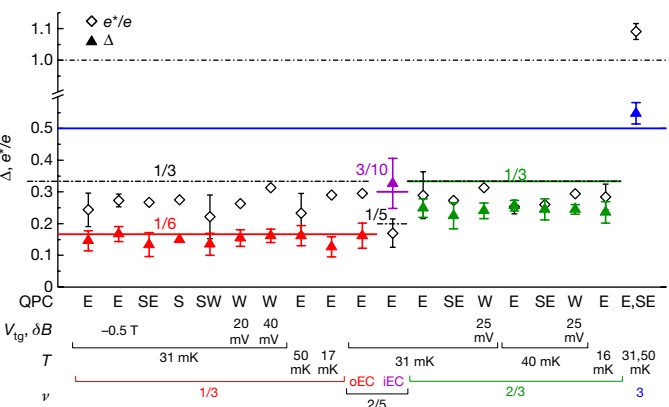

**Fig. 4 | Summary of observations.** Each symbol with error bars represents the mean and standard deviation of the ensemble of $\Delta$ (triangles) and $e^*/e$ (diamonds) extracted along one gate voltage span of a QPC, such as those shown in Fig. 3. The horizontal axis indicates the experimental conditions: label of the QPC (E, SE, S, SW, W), voltage $V_{tg}$ applied to the top gate around QPC$_W$, magnetic field shift $\delta B$ from the centre of the plateau (if any), temperature $T$, filling factor $\nu$ and for $\nu=2/5$ the examined edge channel (inner 'iEC' or outer 'oEC'), with different colours denoting different predicted quasiparticles.

## Conclusion

Fano factor measurements previously used to investigate the charge of tunnelling quasiparticles also allow for a consistent determination of their scaling dimension, from the width and specific shape of $F(eV/k_B T)$. Combined with a systematic approach, the resulting observations of $\Delta$ establish long-lasting theoretical predictions for the fractional quantum Hall quasiparticles at $\nu = 1/3$, 2/5 and 2/3. This approach could be generalized to other quasiparticles, with the potential to shed light on the non-Abelian quasiparticles predicted at even-denominator filling factors. It may also be applied to other low-dimensional conductors.

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

## Methods

### Sample

The sample is nanofabricated by electron-beam lithography on a Ga(Al)As heterojunction forming a 2DEG buried at 140 nm, of density $n = 1.2 \times 10^{11}$ cm$^{-2}$ and of mobility $1.8 \times 10^{6}$ cm$^2$ V$^{-1}$ s$^{-1}$. The 2DEG mesa is first delimited by a wet etching of 105 nm, deeper than the Si $\delta$-doping located 65 nm below the Ga(Al)As surface. The large ohmic contacts (schematically shown as circles in Fig. 1) used to drive and measure the quantum Hall edge currents are then formed 100–200 μm away from the QPCs by electron-beam evaporation of a AuGeNi stack, followed by a 50-s thermal annealing at 440 °C. A 15-nm layer of HfO$_2$ is grown by thermal atomic layer deposition at 100 °C over the entire mesa, to strongly reduce a gate-induced degradation of the 2DEG that could complicate the edge physics. This degradation is generally attributed to unequal thermal contractions on cooling[49] or a deposition stress, which could also modulate the edge potential carrying the quantum Hall channels along the gates. In previous works, we observed a change in the behaviour of QPCs, including in the thermal noise to shot noise crossover, that was correlated with their orientation[28,50] (see also source versus central QPCs in refs. 7,29), which we suspect to result from such gate-induced complication. Here the Ti (5 nm)–Au (20 nm) gates used to form the QPCs are evaporated on top of the HfO$_2$. The five QPCs, of different orientations with respect to the Ga(Al)As crystal, have nominally identical geometries. The split gates have a nominal tip-to-tip distance of 600 nm and a 25° tip-opening angle prolonged until a gate width of 430 nm. Larger-scale electron-beam and optical images of the measured device are shown in Extended Data Fig. 1. The relatively important gate width (about three times the 2DEG depth) was chosen to reduce possible complications from Coulomb interactions between the quantum Hall edges across the gates[9,51] and to better localize the tunnelling location when the QPC is almost open (for less negative gate voltages)[11]. The nominal separation between the split gates controlling QPC$_W$ and the surrounding metal gate is 150 nm. A high-magnification picture of QPC$_W$ with its surrounding metal gate is shown in Extended Data Fig. 2. Note that all the gates were grounded during the cooldown.

### Measurement

The sample is cooled in a cryofree dilution refrigerator and electrically connected through measurement lines both highly filtered and strongly anchored thermally (see ref. 52 for details). Final $RC$ filters with CMS components are positioned within the same metallic enclosure screwed to the mixing chamber that holds the sample: 200 kΩ–100 nF for gate lines, 10 kΩ–100 nF for the bias line and 10 kΩ–1 nF for low-frequency measurement lines. Note a relatively important filtering of the bias line, which prevents an artificial rounding of the thermal noise to shot noise crossover from the flux noise induced by vibrations in a magnetic field. The differential QPC transmission $\partial I_B / \partial I_{inj} = 1 - \partial I_F / \partial I_{inj}$ is measured by standard lock-in techniques at 13 Hz. A particularly small ac modulation is applied on $V$ (of rms amplitude $V_{ac}^{rms} \approx k_B T/3e$) to avoid any discernible rounding of the thermal noise to shot noise crossover. The transmitted and reflected dc currents used to calculate $\tau$ and $F$ are obtained by integrating with the applied bias voltage the corresponding lock-in signal $I_{B,F}(V) = \int_0^V (\partial I_{B,F}/\partial V) dV$.

A specific QPC is individually addressed by completely closing all the other ones. For the composite edges at $\nu = 2/5$ and $\nu = 3$, the characterizing current transmission ratio $\tau$ refers to the current transmission along the specific channel of interest. Explicitly, at $\nu = 2/5$, the transmission $\tau$ along the inner edge channel is given by the ratio between measured (total) $I_B^{meas}$ (only the inner channel of interest is backscattered, the outer channel is fully transmitted, as attested by a broad and noiseless $e^2/3h$ plateau) normalized by the current $Ve^2/15h$ injected along the inner channel: $\tau = I_B^{meas}/(Ve^2/15h)$. For the outer channel at $\nu = 2/5$, the fully backscattered inner edge channel current $Ve^2/15h$

is removed from the measured total $I_B^{meas}$ and the result is normalized by the current $Ve^2/3h$ injected along the outer edge channel: $\tau = (I_B^{meas} - Ve^2/15h)/(Ve^2/3h)$.

Noise measurements are performed using specific cryogenics amplification chains connected to dedicated ohmic contacts, through nearly identical $L$–$C$ tanks of resonant frequency 0.86 MHz (refs. 53,54). The noise ohmic contacts are located upstream of the ohmic contacts used for low-frequency transmission measurements, as shown in Fig. 1. A dc block (2.2 nF) at the input of the $L$–$C$ tanks preserves the low-frequency lock-in signal. For the particular case of QPC$_E$, the forward (transmitted) current fluctuations $\delta I_F$ are also measured, which gives us access to $\langle \delta I_F^2 \rangle$ and to the cross-correlations $\langle \delta I_B \delta I_F \rangle$. Apart increasing the signal-to-noise ratio for QPC$_E$, this allows us to confirm that $\langle \delta I_B^2 \rangle$ matches the more robust cross-correlation signal[55].

The device was immersed in a magnetic field close to the centre of the corresponding Hall resistance plateaus, except when a shift $\delta B$ is specifically indicated. The data at $\nu = 1/3$, $\nu = 2/5$, $\nu = 2/3$ and $\nu = 3$ were obtained at $B = 13.7$ T (13.2 T for $\delta B = -0.5$ T), 11.3 T, 6.8 T and 1.5 T, respectively. See vertical arrows in Extended Data Fig. 3 for the localization of these working points within a magnetic field sweep of the device along $B \in [4, 14]$ T ($\nu \in [1/3, 1]$).

### Thermometry

The electronic temperature inside the device is obtained by the noise measured at thermal equilibrium, with all QPCs closed. For temperatures $T \geq 30$ mK (up to the maximum $T \simeq 55$ mK), we find at $\nu = 1/3$ and $\nu = 3$ that the measured thermal noise is linear with the temperature readings of our calibrated RuO$_2$ thermometer. This establishes the good thermalization of electrons in the device with the mixing chamber, as well as the calibration of the RuO$_2$ thermometer. Accordingly, we indifferently get $T \geq 30$ mK from the equilibrium noise or the equivalent RuO$_2$ readings. At the lowest investigated temperatures of approximately 15 mK, the RuO$_2$ thermometers are no longer reliable and $T$ is obtained from the thermal noise by linearly extrapolating from $S$ ($T \geq 30$ mK). Note that the $S(T)$ slope was not recalibrated at $\nu = 2/3$ but its change from $\nu = 1/3$ was calculated from the separately obtained knowledge of the $L$–$C$ tank circuit parameters, see the next section.

### Noise-amplification chains calibration

The gain factors $G_{F,B}^{eff}$ between raw measurements of the autocorrelations, integrated within a frequency range $[f_{min}, f_{max}]$, and the power spectral density of current fluctuations $\langle \delta I_{F,B}^2 \rangle$ are obtained from:

$$G_{F,B}^{eff} = \frac{s_{F,B}}{4k_B(1/R_{tk} + \nu e^2/h)}, \tag{3}$$

with $R_{tk} \simeq 150$ kΩ the effective parallel resistance accounting for the dissipation in the considered $L$–$C$ tank and $s_{F,B}$ the slope of the raw thermal noise versus temperature. The cross-correlation gain factor is simply given by $G_{FB}^{eff} \simeq \sqrt{G_F^{eff} G_B^{eff}}$, up to a negligible reduction (<0.5%) owing to the small difference between the two $L$–$C$ tanks. In practice, the thermal noise slopes $s_{F,B}$ were only measured at $\nu = 1/3$ and $\nu = 3$. The changes in $G_{F,B}^{eff}(\nu)$ at $\nu \in \{2/3, 2/5\}$ from the gains at $\nu = 1/3$ are obtained from:

$$\frac{G_{F,B}^{eff}(\nu)}{G_{F,B}^{eff}(1/3)} = \frac{\int_{f_{min}}^{f_{max}} \left| Z_{tk}^{-1}(f) + \nu e^2/h \right|^{-2} df}{\int_{f_{min}}^{f_{max}} \left| Z_{tk}^{-1}(f) + e^2/3h \right|^{-2} df}, \tag{4}$$

with the tank impedance given by $Z_{tk}^{-1}(f) = R_{tk}^{-1} + (iL_{tk} 2\pi f)^{-1} + iC_{tk} 2\pi f$, in which $L_{tk} \simeq 250$ μH and $C_{tk} \simeq 135$ pF (see Methods in ref. 50 for details about the calibration of the tank parameters). At $\nu \in \{2/3, 2/5, 1/3\}$, we integrated the noise signal in the same frequency window $f_{min} = 840$ kHz

and $f_{max}$ = 880 kHz. At $\nu = 3$, a larger window $f_{min}$ = 800 kHz and $f_{max}$ = 920 kHz takes advantage of the larger bandwidth of roughly $\nu e^2/2\pi h C_{tk}$.

## Noise tests

Among various experimental checks, we note: (1) the effect of a dc bias voltage on the noise when each of the QPCs are either fully open or fully closed, which is found here to be below our experimental resolution. The present imperceptible 'source' noise could have resulted from poor ohmic contact quality, incomplete electron thermalization in the contacts or dc current heating of the resistive parts of the bias line; (2) the effect of the QPC transmission on the noise at zero dc bias voltage, which is negligible at our experimental resolution. This rules out a possibly higher electron temperature in the ohmic contact connected to the bias line with respect to one connected to a cold ground, which would translate into an increase in $\langle\delta I_B^2\rangle$ at $\tau = 1$ compared with $\tau = 0$. It also shows that the vibration noise in the bias line at frequencies well below 1 MHz does not translate into a discernible broadband excess shot noise for intermediate values of $\tau$.

## Fitting details

The extracted values of $e^*$ and $\Delta$ shown in Fig. 3 and in Extended Data Figs. 7 and 8 represent the best-fit parameters minimizing the variance between the shot-noise data and equation (2). Only the meaningful points are shown and included in the statistical analysis. These fulfil two conditions: (1) an accurate fit of the data can be achieved and (2) the charge does not deviate too much from the expected value. Condition (1) requires a quantitative assessment of the fit accuracy. For this purpose, we used the coefficient of determination $R^2$ and chose to apply the same threshold to all the data taken in similar conditions. Specifically, we automatically discarded fits of $R^2 < 0.9965$ at $\nu = 1/3$ and for the outer channel at $\nu = 2/5$, $R^2 < 0.9966$ for the inner channel at $\nu = 2/5$ and $R^2 < 0.9968$ at $\nu = 2/3$. The number of $S(V)$ sweeps discarded by condition (1) is important, two-thirds of the total number (mostly when $\tau$ is too close to 0 or 1). We checked that the overall results are only marginally affected by the specific threshold value (within reasonable variations). All the points that satisfy condition (1) are shown and included in the statistical analysis of the quasiparticle charge. Condition (2) is subsequently applied to deal with situations in which the fitting charge is found at odds with the predicted value. Specifically, we discarded $S(V)$ sweeps for which the charge is found to be more than 44% off, that is, $e^* < 0.56e_{th}^*$ or $e^* > 1.44e_{th}^*$. The former happens at small $\tau$ with a small QPC gate voltage. This gate voltage might not be enough to deplete the gas under the QPC gates, which could make tunnelling happen in several places along the gates and not only located at their tip, deviating from the model of a point contact. The latter occurs in the so-called strong backscattering regime, in which the nature of the tunnelling quasiparticles is expected to change. Indeed, in the weak backscattering regime ($\tau \ll 1$), the tunnelling barrier between the two edges is made of the electron gas in the fractional quantum Hall regime that selects the quasiparticles. However, in the strong backscattering regime ($\tau \rightarrow 1$), the tunnelling barrier between the two edges is made of vacuum that will select electrons. The points that do not satisfy condition (2) are shown with different symbols and not included in the statistical analysis of $\Delta$. They represent a small fraction (5%) of the data satisfying condition (1).

A complementary fitting procedure was used to further establish the robustness of our results. In Extended Data Fig. 4, we summarize the extracted $\Delta$ obtained by fitting the thermal noise to shot noise crossover of $S(V)$ using for $e^*$ the theoretically predicted value. The fits are performed on the same set of $S(V)$ sweeps as for the main fitting procedure for $\Delta$ (obeying the two above-mentioned conditions (1) and (2)). The voltage bias extension of these fits is reduced to $e^*|V| \leq 2k_B T$ to limit the weight of the shot noise that is only sensitive to $e^*$.

## Predictions

The $\Delta$ predictions indicated in the manuscript for fractional quasiparticles at $\nu = 1/3$ and $2/5$ follow the Luttinger liquid expression $\Delta = (e^*/e)^2/(2Gh/e^2)$ for $e^*$ quasiparticles along a chiral 1D channel of conductance $G$ (refs. 1,13,45). Note that, in these fully chiral states (in which all channels along one edge propagate in the same direction), the quasiparticles exchange phase $\theta$ is predicted to be simply related to $\Delta$ by the relation $\theta = 2\pi\Delta$ (see, for example, Appendix A in ref. 13).

The above Luttinger expression for $\Delta$ does not apply at $\nu = 2/3$ for $e/3$ quasiparticles delocalized between a $2e^2/3h$ edge channel and a neutral counterpropagating channel that further increases $\Delta$; see ref. 56. Note that, in general, the predicted link between $\Delta$ and $\theta$ for fully chiral quantum Hall edges does not hold in the presence of counterpropagating (charged and/or neutral) modes[13].

## Filling factor 2/3

In this more complex hole-conjugate state[1,2,56]: (1) the edges are not fully chiral and found to carry a backward heat current (going upstream to the flow of electricity) and (2) the QPCs can exhibit a plateau at half transmission (see, for example, refs. 46,57). The former may introduce unwanted heat-induced contributions to the noise, whereas the latter alludes to a composite edge structure. Both have possible consequences on the interpretation of the noise signal[58–60].

(1) Non-chiral heating. As in previous works (see, for example, ref. 57), we observe in our device an upstream heating (only) at $\nu = 2/3$, through three noise signatures (see Extended Data Fig. 5). Signature 1: the strongest noise signature, seen at all temperatures (see Extended Data Fig. 5d for $T \simeq 40$ mK), is obtained in the configuration schematically depicted Extended Data Fig. 5a. Here the noise is measured on a contact located electrically upstream a hotspot in an adjacent voltage-biased contact (about 30 µm away, for example, the contact usually used for measuring $\langle I_B\rangle$ in Fig. 1). As shown in Extended Data Fig. 5a, the noise increase is attributed to a local heating of the noise-measurement contact (near the location at which electrical current is emitted from this contact) by the upstream neutral heat current originating from the downstream hotspot. See Fig. 3 of ref. 57 for a similar observation in the same configuration. Note that, in configurations used for investigating $\Delta$, the heat generated at the downstream grounded contacts cannot propagate to the noise-measurement contacts, because a floating contact located in between (measuring $\langle I_{B,F}\rangle$; see Fig. 1) absorbs the upstream heat flow (see Appendix A and Fig. 10 in ref. 58 for a specific discussion). Signature 2: a weaker noise signature from a different heating mechanism is observed, only at the lowest temperature ($T \simeq 17$ mK in Extended Data Fig. 5e), in the configuration schematically depicted Extended Data Fig. 5b (the same configuration is labelled N → C in Fig. 4 of ref. 57). Here a hotspot is created at a downstream contact biased at $V$. The counterpropagating neutral mode carries an upstream heat current to the QPC, which converts the increased temperature into electrical noise from a thermally induced mechanism. The signal is weaker, as would be expected from a smaller heat current through the longer distance of about 150 µm between the hotspot and the QPC (the heat propagation is diffusive owing to thermal equilibration between counterpropagating channels). In practice, it is discernible only at the lowest temperature $T \simeq 17$ mK and for the highest QPC sensitivity ($\tau \approx 0.5$). The lower effect (imperceptible here) at higher temperatures is expected from the generally more efficient relaxation to thermal equilibrium. Note that, in the configurations used to examine $\Delta$, the contacts immediately downstream of the QPC are floating (used to measure the noise or $\langle I_F\rangle$) and, consequently, absorb the upstream heat current originating from the grounded contacts further downstream. Signature 3: a possibly more consequential signature of upstream heating is observed in the same configuration as that used to examine $\Delta$, through an increase in

the noise sum $S_\Sigma \equiv S_F + S_B + 2S_{FB}$. From charge conservation and the chirality of electrical current, $S_\Sigma$ corresponds to the thermal noise emitted from the source contacts electrically upstream of the QPC (independently of any noise generated along the path, such as the partition noise at the QPC, as long as there is no charge accumulation in the investigated MHz range). In fully chiral states such as $\nu \in \{1/3, 2/5, 3\}$, the temperature of the source contacts is independent of the applied bias $V$ (at the emission point) and so is $S_\Sigma$. At $\nu = 2/3$ and $T \simeq 16$ mK, this is not the case, as shown in Extended Data Fig. 5f. This increase in $S_\Sigma$ is interpreted as the signature of a local hotspot in the approximately 150 µm upstream source contacts, by heated-up counterpropagating neutral modes generated at the voltage-biased QPC, as schematically shown in Extended Data Fig. 5c (for a previous observation of the same mechanism, see configuration labelled C → N in Fig. 4 of ref. 57). In practice, we observe a fast increase followed by a near saturation at $S_\Sigma \lesssim 7\,10^{-30}$ A$^2$ Hz$^{-1}$, which is not negligible with respect to the partition noise of interest (see Fig. 2c). To limit the impact of this effect at $T \simeq 16$ mK and $\nu = 2/3$, we only considered the cross-correlation signal ($S \equiv -S_{FB} = -\langle \delta I_F \delta I_B \rangle$) measured on QPC$_E$. Indeed, a symmetric heating of the two source contacts electrically upstream of QPC$_E$ (biased at $V$ and grounded) would not result in any change of the cross-correlations (but instead in a thermally induced increase of the autocorrelations). See ref. 55 for a discussion on the stronger robustness to artefacts of cross-correlations with respect to autocorrelations. At the higher temperatures investigated, there was no discernible change in $S_\Sigma$ and we performed our data analysis using all the noise measurements available.

(2) Noisy $\tau = 1/2$ plateau. A small but discernible 'plateau' is present at $\tau \simeq 1/2$ in the transmission versus split gate voltage of both QPC$_E$ (see inset in Fig. 3c) and QPC$_{SE}$ (see Extended Data Fig. 6). These plateaus, which are robust to the application of a bias voltage $V$ and to temperature changes, suggest the presence of two $e^2/3h$ edge channels sequentially transmitted across the QPC. In that case, there would be no partition noise at the QPC, as observed at $\nu = 3$ and $\nu = 2/5$. By contrast, the small $\tau \simeq 1/2$ 'plateaus' at $\nu = 2/3$ exhibit a substantial noise signal (see also, for example, ref. 46). It was proposed that such noise on a $\tau = 1/2$ plateau was resulting not from the emergence of shot/partition noise but from a heating mechanism involving the thermal equilibration between downstream charge modes and upstream neutral modes[58–60]. In this picture of the QPC at $\tau \approx 0.5$, the fit parameters $e^*$ and $\Delta$ should not be interpreted as the charge and scaling dimension of fractional quasiparticles. Which picture more adequately describes the QPC at $\tau \simeq 1/2$ and $\nu = 2/3$ is not straightforward. On the one hand, whereas the smallness of the $\tau \simeq 1/2$ 'plateaus' does not rule out a simple accidental explanation within the tunnelling picture (from the specific way the barrier deforms with gate voltage, possibly with nearby defects), their mere observation casts doubts on the tunnelling picture and, consequently, on the interpretation of the fit parameters $e^*$ and $\Delta$ near $\tau \simeq 0.5$ as characterizing quantum numbers of fractional quasiparticles. On the other hand, the observation of similar values as for small transmissions, at which the noise signal originates from the tunnelling of fractional quantum Hall quasiparticles across the QPC, suggests that the same tunnelling mechanism is at work at $\tau \simeq 1/2$. In particular, a markedly larger (over-Poissonian) noise would be expected from the heating interpretation in the so-called thermally equilibrated regimes (compared with $e^*/e \approx 0.3$ observed here over a broad transmission range, including $\tau \simeq 1/2$, and theoretically expected for the fractional quantum Hall quasiparticles at $\nu = 2/3$)[58,60]. Overall, more caution is advised on the interpretation of the extracted $e^*$ and $\Delta$ at $\tau \approx 0.5$ for $\nu = 2/3$, compared with lower $\tau$ or different $\nu \in \{1/3, 2/5\}$.

## Extended data

Individual values of $\Delta$ and $e^*/e$ extracted along gate voltage spans are shown in Extended Data Figs. 7 and 8, in complement to Fig. 3.

The dc voltage dependence of the transmission $\tau(V)$ at all gate-voltage tunings in the three configurations shown in Fig. 3 are plotted in Extended Data Fig. 9. Among these are three $\tau(V)$ also shown in the insets of Fig. 2a–c.

## Measured versus tunnelling noise

The measured backscattered current $I_B$ can always be written as the sum $I_B = I_T + I_{gnd}$ of the incident current $I_{gnd}$ emitted from the ohmic contact that would solely contribute to $I_B$ in the absence of tunnelling and of the tunnelling current $I_T$ across the constriction. With this decomposition, the backscattered current noise reads:

$$\langle \delta I_B^2 \rangle = \langle \delta I_T^2 \rangle + 2\langle \delta I_T \delta I_{gnd} \rangle + \langle \delta I_{gnd}^2 \rangle, \tag{5}$$

with $\langle \delta I_{gnd}^2 \rangle = 2k_B T \nu e^2/h$ the thermal noise emitted from the grounded reservoir. Note that because $\langle \delta I_{gnd}^2 \rangle$ is independent of the applied bias voltage $V$, it cancels in the excess noise $S_B$. In the tunnelling limit ($\tau_B \ll 1$), theory predicts from detailed balance between upstream and downstream tunnelling events that the first term in the right side of equation (5) is independent of the scaling dimension $\Delta$ and given by[61]:

$$\langle \delta I_T^2 \rangle = 2e^* \langle I_B \rangle \coth \frac{e^* V}{2k_B T}. \tag{6}$$

The dependence on $\Delta$ of the measured noise thus solely results from the second term on the right side of equation (5), namely, $2\langle \delta I_T \delta I_{gnd} \rangle$. According to the so-called non-equilibrium fluctuation–dissipation relations for chiral systems (assuming a $V$-independent Hamiltonian, as discussed below), this $\Delta$-dependent contribution to the noise is simply given by[62]:

$$\langle \delta I_T \delta I_{gnd} \rangle = -2k_B T \frac{\partial \langle I_B \rangle}{\partial V}. \tag{7}$$

Experimentally, $\partial \langle I_B \rangle/\partial V$ is directly measured. Hence, in this framework, we could calculate the excess noise $S_B^{FDT}$ by plugging the separately measured tunnelling current and its derivative into these equations. This gives (as well as the usual ad hoc correction for large $\tau$)

$$S_B^{FDT} = 2e^*(1-\tau)\langle I_B \rangle \coth \frac{e^* V}{2k_B T} - 4k_B T \left(1 - \frac{\partial \langle I_B \rangle}{\partial I_{inj}}\right) \frac{\partial \langle I_B \rangle}{\partial V}. \tag{8}$$

However, as illustrated in Extended Data Fig. 10, we find that equation (8) using the measured $\langle I_B \rangle(V)$ does not reproduce the simultaneously measured thermal noise to shot noise crossover. This should not come as a surprise as the current and its derivative do not follow Luttinger liquid predictions. One could explain this mismatch by invoking the same possible explanation as for the data–theory discrepancy on the $I$–$V$ characteristics, namely, that the shape of the QPC potential, and therefore the quasiparticle tunnelling amplitude, is affected by external parameters, such as an electrostatic deformation induced by a change in the applied bias voltage, the temperature or the tunnelling current itself[10]. Indeed, as pointed out in ref. 62, equation (7) holds if the voltage bias $V$ only manifests through the chemical potential of the incident edge channel and not if applying $V$ affects the tunnel Hamiltonian.

*Note added in proof:* Coincident to this investigation, two other works are experimentally addressing the scaling dimension of the $e/3$ fractional quantum Hall quasiparticles at $\nu = 1/3$. An experiment by the team of M. Heiblum with a theoretical analysis led by K. Snizhko[63] exploits the same thermal noise to shot noise crossover as in this work, with a focus on low voltages and assuming the predicted fractional charge (see Fig. 4 for such a single-parameter data analysis at low bias), and finds $\Delta \simeq 1/2$. The team of G. Feve (M. Ruelle et al., submitted) relies on a different, dynamical response signature and finds $\Delta \simeq 1/3$. In these two coincident works, the extracted scaling dimension is different from

the pristine prediction $\Delta = 1/6$ observed in this work. As pointed out in the manuscript, the emergence of non-universal behaviours could be related to differences in the geometry of the QPCs.

## Data availability

Further information related to this work is available from the corresponding authors on reasonable request. Plotted data, raw data and data-analysis code are available from Zenodo at https://doi.org/10.5281/zenodo.10599318 (ref. 64).

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

**Acknowledgements** This work was supported by the European Research Council (ERC-2020-SyG-951451) and the French RENATECH network. We thank K. Snizhko for discussions and E. Boulat for providing the τ(*V*, *T*) prediction at *v* = 1/3 in Fig. 2.

**Author contributions** A.V., C.P., P.G., Y.S. and F.P. performed the experiments, with input from A. Aassime and A. Anthore; A.V. and F.P. analysed the data, with input from A. Anthore, C.P., P.G. and Y.S.; A.C. and U.G. grew the 2DEG; A.V., A. Aassime and F.P. fabricated the sample; Y.J. fabricated the high-electron-mobility transistor used in the cryogenic noise amplifiers; A.V. and F.P. wrote the manuscript, with contributions from all authors; A. Anthore and F.P. led the project.

**Competing interests** The authors declare no competing interests.

**Additional information**
**Correspondence and requests for materials** should be addressed to A. Anthore or F. Pierre.

**a**

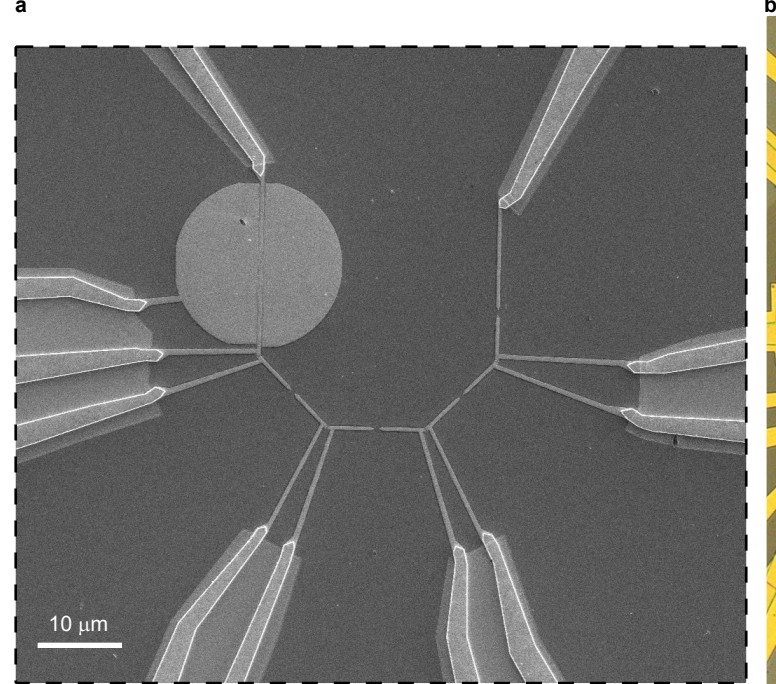

**b**

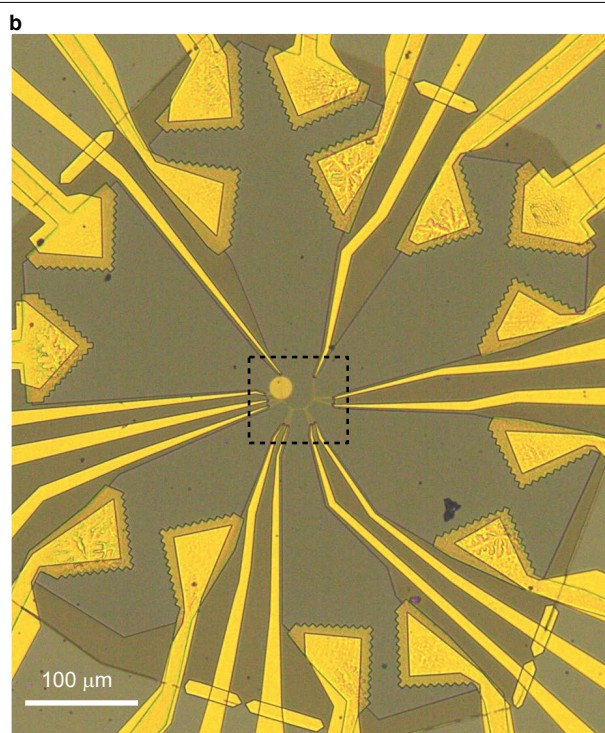

**Extended Data Fig. 1 | Large-scale pictures of the measured device.**
**a**, Electron-beam micrograph. Areas with a 2DEG underneath (the mesa) appear darker. Lighter parts with bright edges are thick, gold layers used to climb down the mesa edges and as bonding pads. **b**, Optical image. The thick top layer made of gold appears as the brightest yellow. Ohmic contacts have staircase edges and show as a darker shade of yellow. The surface over which the HfO$_2$ was deposited (dark grey) completely encapsulates the active part of the device, including ohmic contacts. The dashed rectangle indicates the boundary of the electron-beam image in panel **a**.

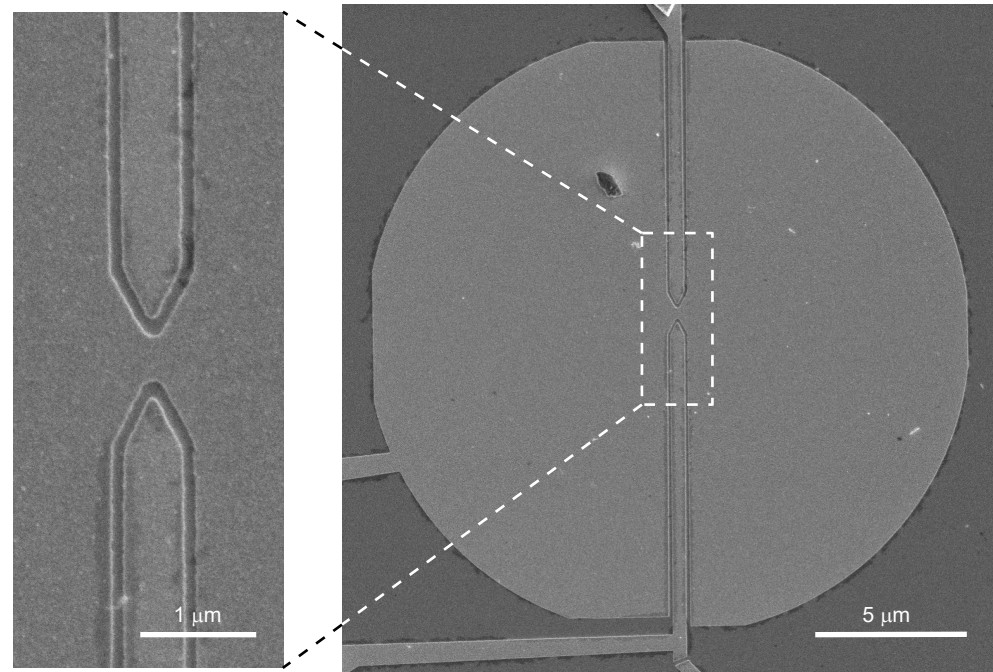

**Extended Data Fig. 2 | QPC with surrounding metal gate.** Electron-beam micrographs of QPC_W.

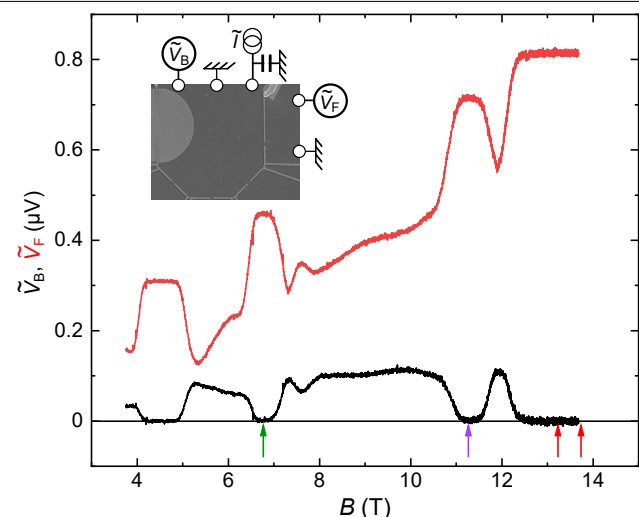

**Extended Data Fig. 3 | Magnetic-field sweep.** Forward ($\tilde{v}_F$, red) and backscattered ($\tilde{v}_B$, black) ac voltages across a fully open QPC$_E$, in response to a fixed ac bias current $\tilde{I}$, are plotted as a function of magnetic field $B$. The other QPCs are fully closed. Arrows indicate at which $B$ the different measurements were performed (except $B \simeq 1.5$ T at $v = 3$, not shown here). At these points, the backscattered signal is zero, whereas the forward signal is well within a plateau, despite the increased mixing-chamber temperature of 160 mK during this $B$ sweep. Note that $\tilde{v}_F$ does not precisely scale as $h/ve^2$ along plateaus, owing to the parallel capacitance shown schematically (100 nF) and the finite ac frequency (13 Hz).

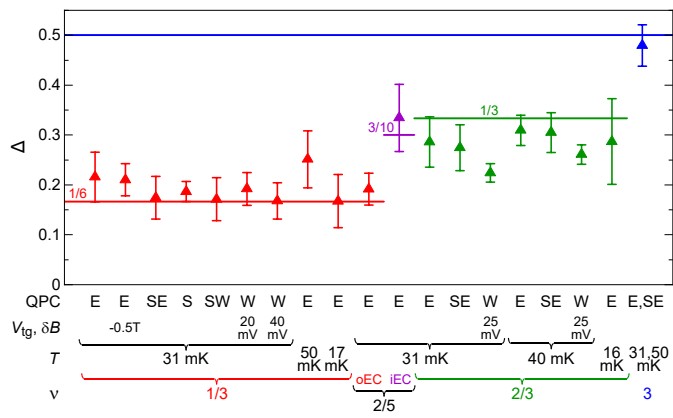

**Extended Data Fig. 4 | Summary of single-parameter analysis.** Symbols recapitulate the extracted scaling dimension in all explored configurations, similarly to Fig. 4 but with Δ obtained by a different procedure. The quasiparticle charge $e^*$ is here assumed to take its predicted value and the fit is performed only on the thermal noise to shot noise crossover at low bias and using Δ as the only free parameter (see text).

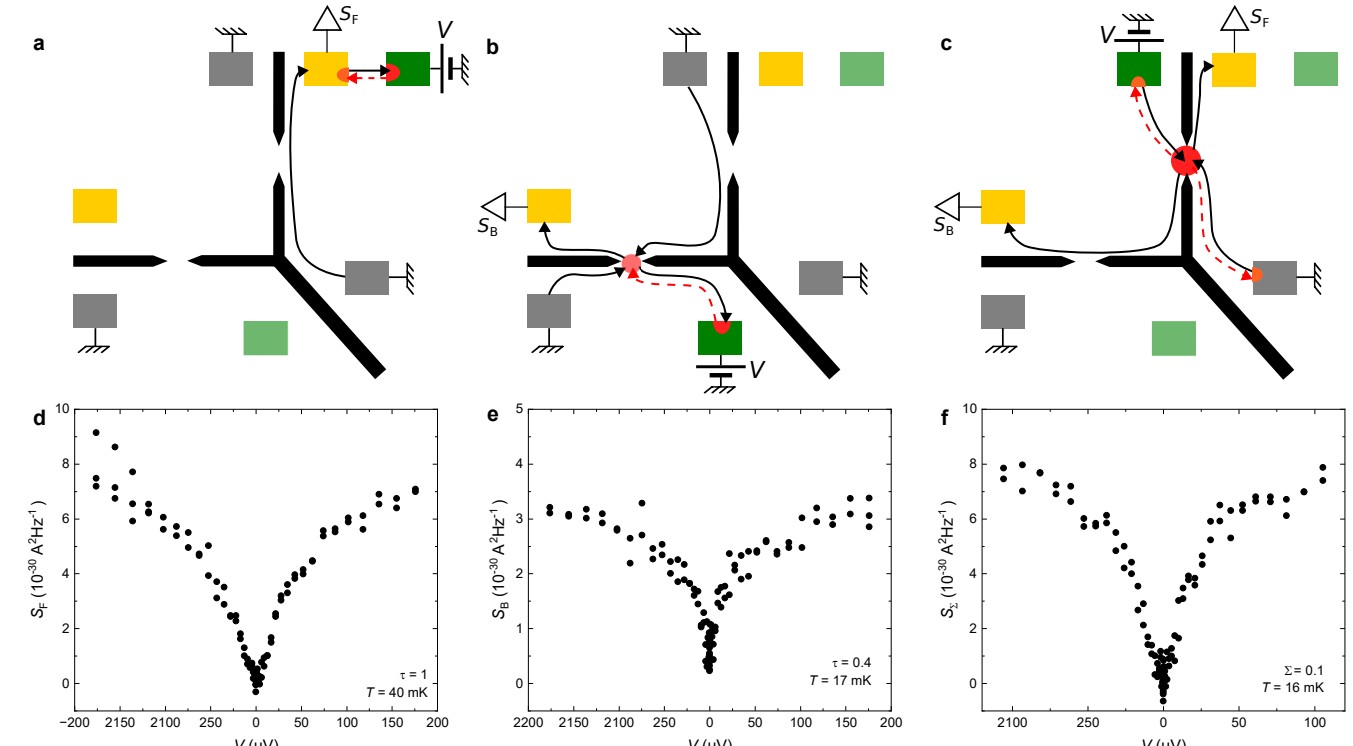

**Extended Data Fig. 5 | Signatures of upstream neutral heat flow at ν = 2/3.** The presence of a neutral heat current flowing in the opposite direction of the electrical current is assessed by noise measurements in three different configurations. **a**–**c**, Schematic illustrations of the three processes (described in the text) by which heat is created, transported upstream by the neutral modes (dashed red arrows) and detected. **d**–**f**, Noise signature of upstream heating measured in the configuration shown in the panel immediately above.

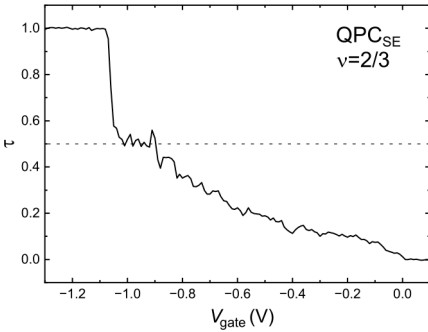

**Extended Data Fig. 6 | Transmission 'plateau' across QPC$_{SE}$ at $v$ = 2/3.** The measured QPC 'backscattering' transmission $\tau$ across QPC$_{SE}$ at $v$ = 2/3 is plotted as a continuous line versus gate voltage $V_{gate}$. The horizontal dashed line indicates $\tau$ = 1/2. See inset in Fig. 3c for the corresponding gate-voltage sweep of QPC$_E$ at $v$ = 2/3.

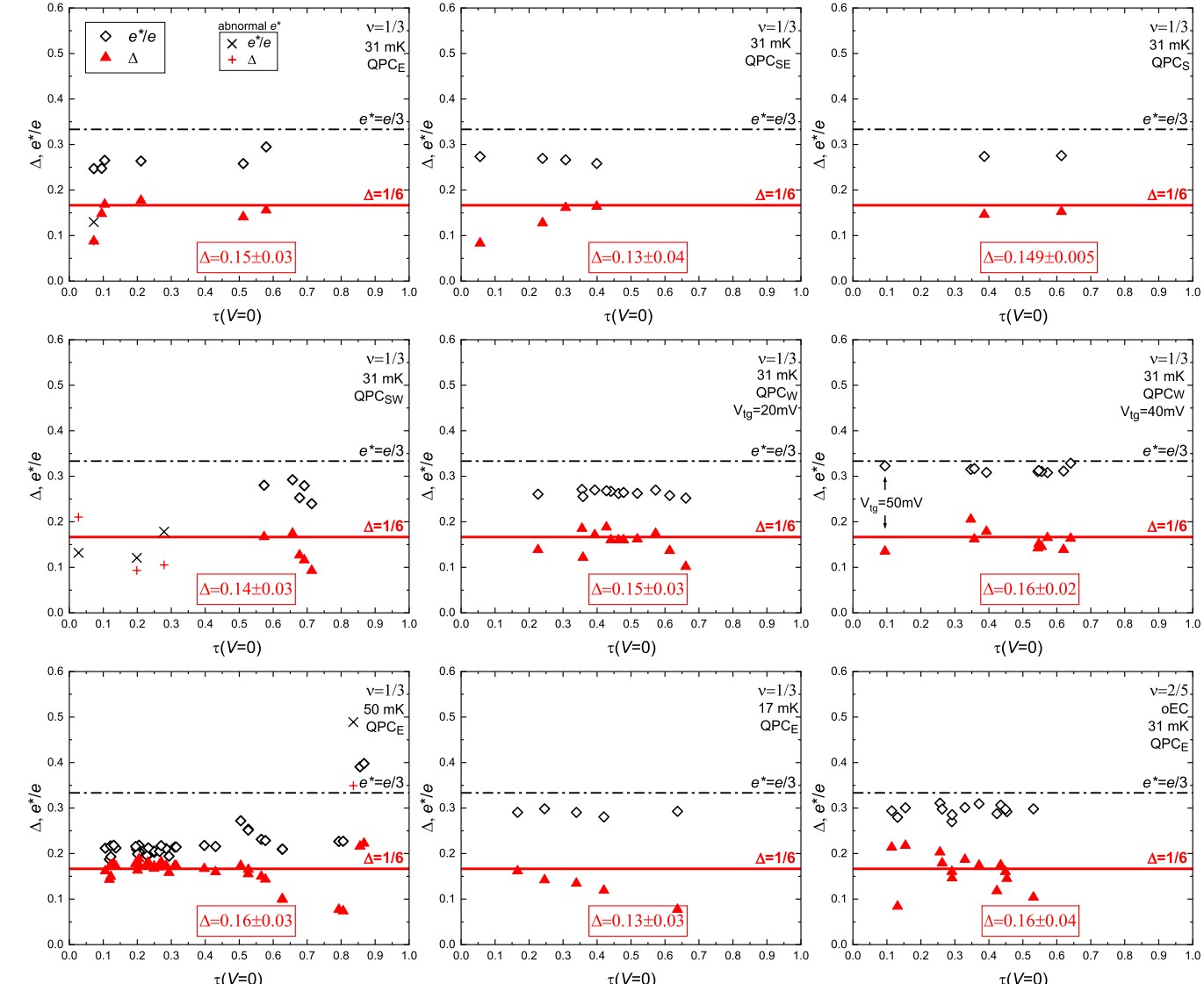

**Extended Data Fig. 7 | Scaling dimension versus QPC tuning of predicted {$e$/3, Δ = 1/6} quasiparticles.** Individual values of extracted scaling dimension (triangles) and charge (diamonds) are plotted versus $\tau(V = 0)$ for each configuration addressing the predicted {$e$/3, Δ = 1/6} fractional quantum Hall quasiparticles. A few points associated with anomalously low or high charge are shown as different symbols (+, ×). Each panel corresponds to the configuration indicated within it. The average and spread of Δ indicated in the panels are calculated only on points shown as triangles and correspond to the individual symbols with error bars in Fig. 4. All measurements are at $\nu = 1/3$ except the bottom-right panel addressing the outer edge channel at $\nu = 2/5$. The configuration corresponding to {$\nu = 1/3$, QPC$_E$, 31 mK, $\delta B \simeq -0.5$ T} is shown in Fig. 3a.

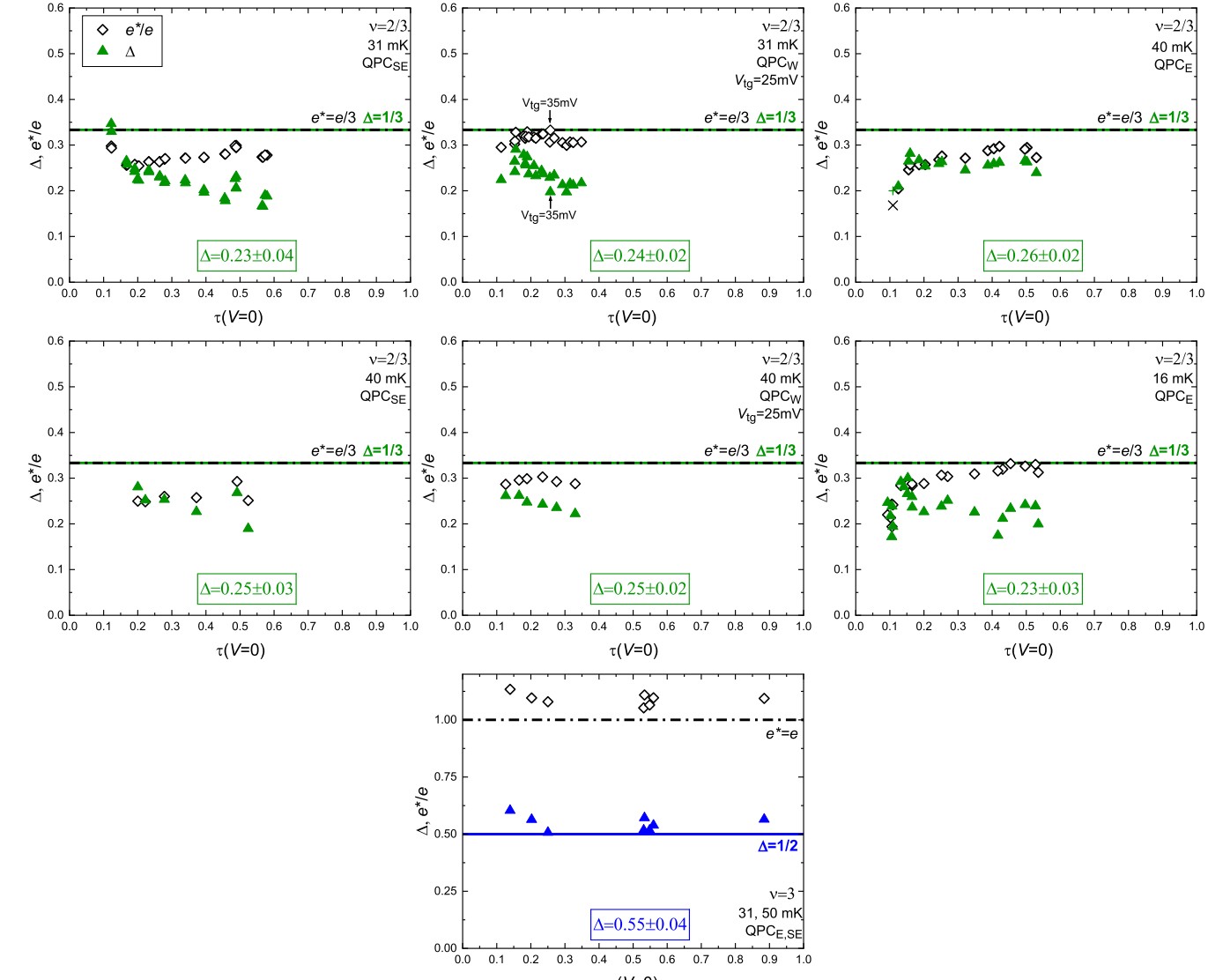

**Extended Data Fig. 8 | Scaling dimension versus QPC tuning at ν = 2/3 and ν = 3.** Individual values of extracted scaling dimension Δ (triangles) and charge $e*/e$ (diamonds) are plotted versus $τ(V=0)$. A few points associated with anomalously low or high charge are shown as different symbols (+, ×). Each panel corresponds to the configuration indicated within it. The average and spread of Δ indicated in the panels are calculated only on points shown as triangles and correspond to the individual symbols with error bars in Fig. 4. The upper six panels correspond to $v = 2/3$, whereas the bottom panel corresponds to $v = 3$. The configuration corresponding to $\{v = 2/3, \text{QPC}_E, 31\text{ mK}\}$ is shown in Fig. 3c.

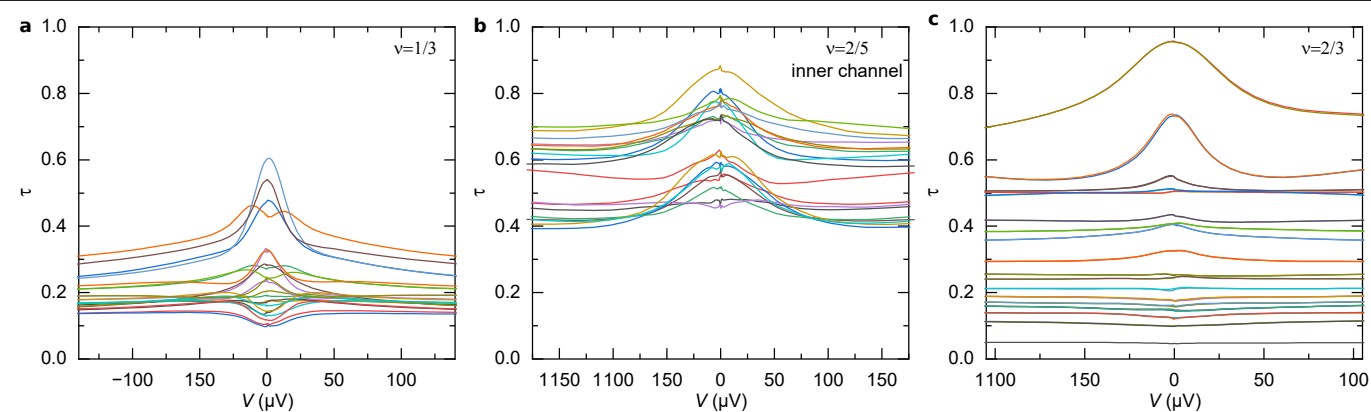

**Extended Data Fig. 9 | Transmission versus dc voltage bias at different gate voltages.** The measured QPC 'backscattering' transmission $\tau$ is plotted versus $V$ for the different gate voltage-tunings and three $\mathrm{QPC_E}$ configurations shown in Fig. 3. Each individual tuning in each panel is shown as a line of different colour. Panels **a**–**c** correspond to the $v = 1/3$, 2/5 inner channel and 2/3 configurations shown in Fig. 3a–c, respectively.

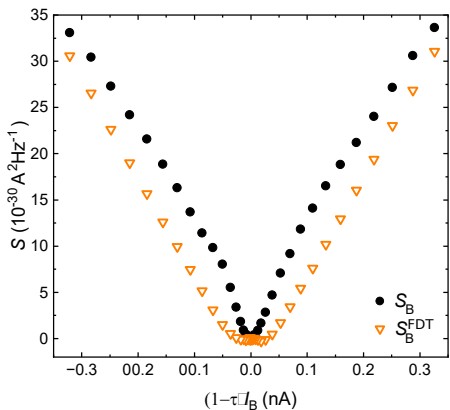

**Extended Data Fig. 10 | Measured $S_B$ versus calculated $S_B^{FDT}$.** Illustrative comparison at $\nu = 1/3$ between the measured excess noise $S_B$ and the value $S_B^{FDT}$ calculated from equation (8) (derived with the non-equilibrium fluctuation–dissipation relation in equation (7)) using the simultaneously measured $\langle I_B \rangle(V)$. The noise shown here is the same as in Fig. 2a. The mismatch could be explained by the same mechanism invoked for the $I$–$V$ characteristics (see Methods).