## [Peer Review File · Nature]

Manuscript Title: Observation of the scaling dimension of fractional quantum Hall anyons

Reviewer Comments & Author Rebuttals

Reviewer Reports on the Initial Version:

Referees' comments:

Referee #1 (Remarks to the Author):

In this manuscript, the authors successfully observe the scaling dimension associated with fractional charge transport, which has not been measured for a long time, using current noise measurements. To confirm reproducibility, measurements were performed at multiple QPCs and filling factors in order to precisely verify the experimental results. The manuscript is clear and well written.

However, I believe that these precise measurements are not as impactful as the previous experiments on fractional charge and anyonic phase. Experiments demonstrating the existence of fractional charge [R. de Picciotto et al., *Nature* 389, 162 (1997), L. Saminadayar et al., *PRL* 79, 2526 (1997)] and the existence of anyonic phase [J. Nakamura et al., *Nat. Phys.* 16, 931 (2020), H. Bartolomei et al., *Science* 368, 173 (2020)] have had a significant impact not only on the fractional quantum Hall field but also on other condensed matter fields. For example, with regard to anyons, the experimental results showing that quasiparticles in semiconductor solids have anyonic statistics had a great impact on researchers working with systems searching for Majorana fermions, such as topological superconductors and superconducting junction systems, because Majorana particles also have anyonic statistics.

On the other hand, I do not believe that the results of this scaling dimension observation will have that much impact. I understand that the result is theoretically important and is significant for analysis on FQH experiments, but I do not know if it is important for experimental studies in other fields. Therefore, I am opposed of publication in *Nature*, and instead I recommend that the results be published in a specialized journal.

I have some questions:

1. The authors focus on the crossover region at low bias voltage. Is it possible in principle to observe scaling dimension in other samples treated in the previous noise measurements [R. de Picciotto et al., *Nature* 389, 162 (1997), L. Saminadayar et al., *PRL* 79, 2526 (1997)], if they perform the same measurement and analysis? Or do they need to do some other things at the sample preparation stage?
2. Could you tell me why $\Delta=1/6$ when the outer channel at $\nu=2/5$ and $\Delta=3/10$ when the inner channel at $\nu=2/5$?
3. Is there an intuitive explanation as to why the noise increases in the thermal noise dominated

region as Δ decreases?

4. I would like to know the meaning of “non-canonical” QPC. What is the difference from a common QPC?

Referee #2 (Remarks to the Author):

It has been of particular interest to measure the scaling dimension Δ of fractional quantum Hall anyons in a quantum point contact setup. Although this quantity is non-universal depending on many details including the interaction between counter-propagating edge modes, the knowledge of Δ can be used for understanding transport observables in the recent experimental activities for anyon braiding. The authors have shown that Δ can be extracted from the thermal noise to shot noise crossover as recently proposed theoretical predictions of Refs. [11, 12]. On the other hand, the I-V curve from which experimentalists traditionally seek to obtain Δ , does not fit to extract Δ properly.

I found the results of paper interesting and scientifically solid. Particularly, I really appreciate their systematic analysis through repeated experiments on different conditions. However, I am not convinced that the above finding reaches the very stringent criteria for publication in Nature. In contrast with topological properties of anyonic excitations (charge and braiding statistics), Δ is a non-universal quantity which may vary in different samples. Indeed, as noted at the end in the paper, the multiple teams find different values of Δ at the same filling $\nu=1/3$. I believe that such a non-universal quantity in this topological material is less attractive to a broad range of readers. But, given that this study may be useful for further anyon braiding-related experiments, I recommend the paper in a more specialized journal, e.g., Nature physics, if the authors address the following questions:

1. I do not agree that the noise on the $\tau=1/2$ plateau at the filling factor $\nu=2/3$ is of the same origin as the usual partitioning noise in the Poissonian limit. As noted in the paper as well, the noise mechanism involves heating due to equilibration between the modes. The application of bias voltage heats the system and thus the elevated temperature leads to noise. Although this noise is linear at high bias voltage like the usual shot noise, the obtained Fano factor is unrelated to the charge of transmitted particles, e.g., see Refs. [61, 63] for more details. On similar grounds, I am not sure that the obtained Δ involves the scaling dimension of anyons. Indeed, the obtained values of Δ rather deviates from the predicted value, compared with the remarkable agreement in the other filling factors.

2. The measured Fano factor at the filling factor $\nu=2/3$ is close to $1/3$, which is inconsistent with the previous results $F = 2/3$, e.g., Ref. [60]. Can you make comments on this?

Minor points:

1. I could not find the data point in Fig. 3, which corresponds to Fig. 2d.
2. I think “Note that in configurations used for investigating Δ , the noise measurement contacts are

electrically upstream a floating contact (used to measure $\langle IB, F \rangle$) and, consequently, there is no such hot spot” is rather misleading. If I understand correctly, e.g., in Fig. 1, the ground contact where I_B enters provides a hot spot. But the generated heat cannot propagate to the noise measurement contact since the floating contact in the middle absorbs the heat.

3. While I found the paragraph “(i) non-chiral heating” in Methods section very interesting, the present texts are not transparent.

Referee #3 (Remarks to the Author):

A. The authors report on measurements of low bias noise observed in quantum point contacts operated in the fractional quantum Hall regime. Their qpcs are fabricated on a two-dimensional electron gas residing in an AlGaAs/GaAs heterostructure. They extract the scaling dimension Δ for $\nu=1/3$, $\nu=2/5$, and $\nu=2/3$ by examining the crossover from thermal noise to shot noise via the change in the Fano factor.

B. The measurement of noise generated by partitioning edge states at qpcs in the fractional quantum Hall regime is not new - there are many reports in the literature. However the authors do appear to focus their analysis on new and interesting regime associated with the transition from thermal noise to shot noise at modest bias with the aim to extract the scaling dimension at each fraction. This aspect is certainly original and significant. Furthermore, the determination of the scaling dimension has taken on new importance in the study of anyons in the fractional quantum Hall regime. By this measure, the research is timely, original, and potentially highly impactful.

C. The data is of high quality and generated by established experts in the field. I do not doubt the veracity of their experimental results. I do request that the authors add some details for clarification and simplicity of understanding. For example, the authors should clearly state the definition of τ in equation 1 and equation 2. This is known to experts, but it should nevertheless be stated explicitly once in the main text. Secondly, the authors should mention the error bars on their experimental determination of noise as a function of bias. It may be that the error bars are smaller than the points used to display the data, but it is never stated explicitly. Just showing data points without an error estimate implies perfect precision. Since precision knowledge of electron temperature in the device is also crucial, it would also be nice explicitly state the electron temperature for the data in Figure 2 rather than saying $T \approx 30\text{mK}$ in passing in the text. If the temperature is fixed at $T=31\text{mK}$, please include this information in the caption or directly in the figure.

D. The use of statistics seems reasonable. Some of the uncertainties associated with the analysis may require elaboration. I discuss further in my subsequent comments.

E. The data and analysis is intriguing and suggestive that the authors have a reasonable method to measure the scaling dimension Δ in the fractional states. It also raises some questions. The qpc transmission vs. gate voltage at $1/3$ is full of resonances, forcing the authors to exclude certain points from the determination of the scaling dimension. This may be reasonable in this specific case, but it does uncertainty to the methodology. For example, in Fig. 3a the extracted charge is

systematically low as the transmission is varied. Can the authors associate this observation with any systematic effect? The scaling dimension is also evaluated over a large range of τ while strictly speaking equation 2 is only valid with $\tau \ll 1$. Can the authors comment on why extraction of Δ over such a broad range is justified? While the extracted Δ at $1/3$ filling factor is impressively close to the theoretical value of $1/6$, much more variation is observed at $\nu=2/5$ and $\nu=2/3$. At these fractions the extractions of the scaling dimension appear less robust. It would be helpful if the authors could comment on this observation. Notably, both $2/3$ and $2/5$ support multiple edge modes. Do interactions between the modes potentially lead to corrections to the scaling dimension? Figure 4 is a nice summary of their findings over multiple qpcs and under different settings.

F. The claim of the paper is that the scaling dimension is determined unambiguously for several fractional states. While the data and analysis support this conclusion for the $1/3$ state, the data for $2/5$ and $2/3$ suggest the complications due to disorder and nonuniversal physics still play a role in these measurements. The manuscript would be strengthened if the authors discuss what is necessary to remove remaining non-idealities in the measurements and/or samples. The manuscript is valuable for its discussion of a new approach to determination of the scaling dimension. It will be even stronger if the authors can suggest steps necessary to make the technique even more reliable.

G. The references are appropriate.

H. The paper is well written and discussion of data collection and analysis is clear. This is a valuable contribution to the field. My main suggestion is more discussion of why the analysis may deviate from theoretical expectations in specific cases.

Author Rebuttals to Initial Comments:

We would like to express our appreciation for the referees' in depth remarks. We believe the manuscript is now improved thanks to the referees' suggestions, comments and questions.

The most important criticism from Referee #1 and #2 concerns the importance and potential impact of the paper. We wish to briefly summarize here our responses on this point developed below. We would like to stress first that the scaling dimension is a characterizing quasiparticle quantum number that broadly applies, well-beyond the fractional quantum Hall regimes. Moreover, the consequences of a non-trivial scaling dimension are far reaching, arguably rivaling those of a fractional charge and quantum statistics. Even if the nature of quasiparticles, such as their scaling dimension, can be impacted by non-universal phenomena, we here manage to observe the pristine fractional quantum Hall predictions. Finally, we anticipate that the novel experimental approach demonstrated here in the fractional quantum Hall regime could be applied to a much broader range of systems.

Referee #1

In this manuscript, the authors successfully observe the scaling dimension associated with fractional charge transport, which has not been measured for a long time, using current noise measurements. To confirm reproducibility, measurements were performed at multiple QPCs and filling factors in order to precisely verify the experimental results. The manuscript is clear and well written.

However, I believe that these precise measurements are not as impactful as the previous experiments on fractional charge and anyonic phase. Experiments demonstrating the existence of fractional charge [R. de Picciotto et al., *Nature* 389, 162 (1997), L. Saminadayar et al., *PRL* 79, 2526 (1997)] and the existence of anyonic phase [J. Nakamura et al., *Nat. Phys.* 16, 931 (2020), H. Bartolomei et al., *Science* 368, 173 (2020)] have had a significant impact not only on the fractional quantum Hall field but also on other condensed matter fields. For example, with regard to anyons, the experimental results showing that quasiparticles in semiconductor solids have anyonic statistics had a great impact on researchers working with systems searching for Majorana fermions, such as topological superconductors and superconducting junction systems, because Majorana particles also have anyonic statistics.

On the other hand, I do not believe that the results of this scaling dimension observation will have that much impact. I understand that the result is theoretically important and is significant for analysis on FQH experiments, but I do not know if it is important for experimental studies in other fields. Therefore, I am opposed of publication in *Nature*, and instead I recommend that the results be published in a specialized journal.

[Reply]

A central characteristics of exotic quasiparticles is their peculiar dynamics, with important consequences (see the first answer to Referee #2), and this is quantified by the scaling dimension.

This quantum number is not limited to the quantum Hall regime, but it extends to a large variety of quasiparticles and, in particular, to those emerging in topological insulators or in 1D systems.

The importance of measuring the scaling dimension of quasiparticles is further reinforced by the physics community's awareness of the considerable challenge of establishing unambiguously their exotic nature. One timely illustration is provided by the debated observation of Majorana modes in hybrid nanowires for protected topological quantum processing.

The present robust measurement strategy based on the thermal-shot noise crossover is not limited to the fractional quantum Hall regimes, but it could be extended beyond this paradigm, and in particular to all 1D conductors generically described by the Luttinger theory.

We hope that these arguments, as well as the complementary arguments in our first answer to Referee #2, will be favorably received by the referee.

In response to the referee, we explicitly state in the conclusion that the present approach could be applied to other systems than the present fractional quantum Hall paradigm.

[Referee #1]

I have some questions:

1. The authors focus on the crossover region at low bias voltage. Is it possible in principle to observe scaling dimension in other samples treated in the previous noise measurements [R. de Picciotto et al., Nature 389, 162 (1997), L. Saminadayar et al., PRL 79, 2526 (1997)], if they perform the same measurement and analysis? Or do they need to do some other things at the sample preparation stage?

[Reply]

Regarding sample preparation: We previously observed different behaviors including thermal to shot noise crossover depending on QPC orientation vs crystal lattice, in devices where the gates are directly deposited at the surface of the AlGaAs heterostructure [29] (also seen by the team of G. Fève in his 'collider' experiments [6,30]). We attribute this phenomenon to the stress induced by the stronger thermal contraction of the metal gates (which is a phenomenon better known in Si devices, where it can give rise to unwanted quantum dots). Reliable and highly reproducible results were obtained here by intercalating an ALD grown HfO₂ layer (of similarly small thermal contraction as AlGaAs) between metal and AlGaAs heterostructure. There are only few very recent experiments using such HfO₂ underlayer. In particular, it is not the case of the pioneer works mentioned by the referee.

Regarding previous noise measurements: Observing the scaling dimension from the thermal to shot noise crossover is technically challenging. It involves simultaneously fulfilling several requirements. (i) The instrumental noise resolution must be particularly good since there is less noise signal at low bias. It is not sufficient in the mentioned pioneers works, where the resolution is within $1-3 \cdot 10^{-30} \text{ A}^2/\text{Hz}$ compared to $1 \cdot 10^{-31} \text{ A}^2/\text{Hz}$ here. (ii) The electronic temperature T in the device must be independently and relatively accurately determined as T directly affects the width of the crossover. This can be challenging whereas it is generally unnecessary in experiments focused on the shot noise determination of the charge. For instance, T is not independently measured at $\nu=1/3$ in the pioneer work of de Picciotto et al. (iii) The voltage bias increment must be sufficiently small compared to the width of the crossover (of the order of kT/e). This is not the case for many experiments focused on the shot noise at large bias, such as the pioneer work of Saminadayar et al.

Consequently, previous noise measurements were generally not well suited to reliably extract the pristine scaling dimension of fractional quantum Hall particles, which is observed here.

In response to the referee, we have added in Methods (subsection 'Sample') a comment regarding the possible orientation dependent impact of the metallic split gate on the QPC behavior.

[Referee #1]

2. Could you tell me why $\Delta=1/6$ when the outer channel at $\nu=2/5$ and $\Delta=3/10$ when the inner channel at $\nu=2/5$?

[Reply]

Chiral Luttinger liquid theory generally predicts that a quasiparticle of charge e^* along a chiral edge channel of conductance G has a scaling dimension $\Delta=(e^*/e)^2/(2G h/e^2)$ and an anyonic exchange phase of $\theta=2\pi \Delta$ (see eg Appendix A, section 1 in [12]).

For tunneling quasiparticles of $e/3$ charge along the outer channel of conductance $e^2/3h$, this gives $\Delta=1/6$ and $\theta=\pi/3$ (the same as at $\nu=1/3$, where quasiparticles of identical $e/3$ charge propagate along a single channel of identical $e^2/3h$ conductance).

For tunneling quasiparticles of $e/5$ charge along the inner channel of conductance $e^2/15h$, this gives $\Delta=3/10$ and $\theta=3\pi/5$.

The above expression for the scaling dimension does not apply for tunneling $e/3$ quasiparticles at $\nu=2/3$. This is because these quasiparticles are delocalized over a charged edge channel and a neutral edge channel propagating in opposite directions (see Eq. 16 in [50], the left term of Eq. 16 associated with the charged edge channel corresponds to the above expression for Δ with $e^*=e/3$ and $G h/e^2=2/3$, the right term is the contribution of the neutral edge channel).

In response to the referee, we have added at the end of the section entitled “Characterizing exotic quasiparticles” the corresponding channel conductance, more specific references to the predictions, and we point to the new subsection ‘Predictions’ in Methods where the above Luttinger expression is provided together with a discussion of the specific $2/3$ case.

[Referee #1]

3. Is there an intuitive explanation as to why the noise increases in the thermal noise dominated region as Δ decreases?

[Reply]

Our intuitive picture is to connect the longer autocorrelation time for quasiparticles of lower Δ (autocorrelations decrease as $1/t^2\Delta$) to a lower characteristic energy scale e^*V over which the crossover develops. In this picture, the shot noise triggers at lower bias above thermal noise for lower Δ , which results in more noise.

In response to the referee, we now provide this intuitive picture in the manuscript, below Eq. 2.

[Referee #1]

4. I would like to know the meaning of “non-canonical” QPC. What is the difference from a common QPC?

[Reply]

We were referring to non-ideal behaviors of QPCs, a straightforward example of which is the emergence of a quantum dot behavior due to nearby impurities.

In response to the referee, we have changed the word ‘canonical’ by ‘ideal’, and we point out the presence of localized electronic levels as an example of non-idealities.

Referee #2

It has been of particular interest to measure the scaling dimension Δ of fractional quantum Hall anyons in a quantum point contact setup. Although this quantity is non-universal depending on many details including the interaction between counter-propagating edge modes, the knowledge of Δ can be used for understanding transport observables in the recent experimental activities for anyon braiding. The authors have shown that Δ can be extracted from the thermal noise to shot noise crossover as recently proposed theoretical predictions of Refs. [11, 12]. On the other hand, the I-V curve from which experimentalists traditionally seek to obtain Δ , does not fit to extract Δ properly.

I found the results of paper interesting and scientifically solid. Particularly, I really appreciate their systematic analysis through repeated experiments on different conditions. However, I am not convinced that the above finding reaches the very stringent criteria for publication in Nature. In contrast with topological properties of anyonic excitations (charge and braiding statistics), Δ is a non-universal quantity which may vary in different samples. Indeed, as noted at the end in the paper, the multiple teams find different values of Δ at the same filling $\nu=1/3$. I believe that such a non-universal quantity in this topological material is less attractive to a broad range of readers. But, given that this study may be useful for further anyon braiding-related experiments, I recommend the paper in a more specialized journal, e.g., Nature physics, if the authors address the following questions:

[Reply]

We would like to stress that the present work demonstrates a robust agreement with pristine theoretical predictions, free of non-universal corrections, thanks to a specific sample design and preparation.

We agree that, in general, the quasiparticles scaling dimension, and therefore the quasiparticles themselves, could be impacted by e.g. interaction between counter propagating channels. Note however that this would also be the case of the charge and statistics of 1D quasiparticles along such interacting edges (as illustrated with e.g. the charge fractionalization along nearby, counter propagating integer quantum Hall edges observed in Kamata et al., Nature Nanotechnology 9, 177 (2014)). Moreover, even when different quasiparticles emerge from additional interaction mechanisms, they remain characterized by their scaling dimension.

We also would like to emphasize that, although less immediately striking than a fractional charge and quantum statistics, the scaling dimension is arguably as central to the quasiparticles' characteristics:

The scaling dimension of quasiparticles determines their dynamical behavior, which is crucial in the context of time-resolved manipulation and in particular for envisioned protected quantum information processing in the non-abelian case.

The scaling dimension has a major influence on the tunneling of quasiparticles, and thereby on most of their transport properties. As pointed out by the referee, the knowledge of Δ is notably crucial to the aim of investigating quantitatively the quasiparticles quantum statistics in 'collider' setups.

The knowledge of the scaling dimension is considered to be essential for differentiating between several possible types of quasiparticles, such as at $\nu=5/2$ in the fractional quantum Hall regime.

We hope that these arguments, as well as the complementary arguments in our first answer to Referee #1, will be favorably received by the referee.

[Referee #2]

1. I do not agree that the noise on the $\tau=1/2$ plateau at the filling factor $\nu=2/3$ is of the same origin as the usual partitioning noise in the Poissonian limit. As noted in the paper as well, the noise mechanism involves heating due to equilibration between the modes. The application of bias voltage heats the system and thus the elevated temperature leads to noise. Although this noise is linear at high bias voltage like the usual shot noise, the obtained Fano factor is unrelated to the charge of transmitted particles, e.g., see Refs. [61, 63] for more details. On similar grounds, I am not sure that the obtained Δ involves the scaling dimension of anyons. Indeed, the obtained values of Δ rather deviates from the predicted value, compared with the remarkable agreement in the other filling factors.

[Reply]

The referee's interpretation of the noise at $\tau=1/2$ for $\nu=2/3$ is also related to the extra complexity of this state hosting counter-propagating modes and whose edge structure remains debated.

We would like to point out several issues with this heat-induced noise interpretation of the $\tau=1/2$ data in our device, taking Table I of [Manna, Das, Goldstein, arXiv:2307.05173v2] as the summary of theoretical expectations:

- Except at $T=16\text{mK}$, we observe the same auto and cross correlations when these different measurements are possible (ie on QPC_E, up to the negative sign of cross-correlations). According to the above mentioned summary table, this would be the case in the unequilibrated stochastic regime. However, in this situation, a value of $F=2/3$ is expected at high bias, whereas we find $F\sim 1/3$ (as also observed by the team of G. Fève at $\tau=1/2$ for $\nu=2/3$ in [26]).
- At 16mK , as indicated in the Methods subsection discussing the experimental challenges of the $2/3$ state, the cross-correlations signal remains $|F|\sim 1/3$ at high bias whereas a noticeable increase in the auto-correlations develops (which we attribute to a non-negligible upstream heating emerging due to an increased equilibration length). No configurations match such observation in the above mentioned summary table.
- Qualitatively, the observed weak dependence on τ and on T appears to be inconsistent with the heating picture at $\tau=1/2$ (and to an important heating contribution away from $\tau=1/2$, in addition to shot noise).

Instead we find that our observations are reasonably consistent with the most widespread picture from Kane, Fisher and Polchinski of a downstream charge mode and an upstream neutral mode [50], despite the emergence of a (here rather small) plateau at $\tau=1/2$ that does not naturally develop in this edge structure picture. In particular, F at high bias (the fitted fractional charge) of ~ 0.3 and the fitted scaling dimension of ~ 0.25 are both reasonably close to the $1/3$ prediction from Kane, Fischer and Polchinski. In addition, these two parameters are remarkably stable for a broad range of τ (see Fig. 3c and Supplementary Fig. S5).

Note that we do not exclude that a more complex edge structure could account for the small but discernible discrepancy between theory and experiment on Δ (see also Fig. 5 for a data analysis assuming the expected e^* , where the difference on Δ between observation and theory is mostly within error bars at $\nu=2/3$).

In response to the referee, we specifically point out alternative edge structures for the $2/3$ states at the end of the section 'Robustness of observations', and much improved the corresponding $2/3$ discussion in Methods.

[Referee #2]

2. The measured Fano factor at the filling factor $\nu=2/3$ is close to $1/3$, which is inconsistent with the previous results $F = 2/3$, e.g., Ref. [60]. Can you make comments on this?

[Reply]

To our knowledge, $F \sim 1/3$ is observed by a majority of teams, including in the present work.

In [49] ([60] in previous article version), autocorrelations measurements of the shot noise correspond to $F=2/3$ at low temperatures and to $F=1/3$ at higher temperatures (at least at $\tau=1/2$ for which the temperature dependence is shown).

In [26], the team of Gwendal Fève found $F \sim 1/3$ from standard shot noise measurements (at 50mK, over a broad range of $\tau < 0.6$ including on a noisy plateau at $\tau=0.5$, see also Supplementary Fig 2 in the separate Supplementary Information file of this reference) and the corresponding $e/3$ charge (in the shot noise interpretation) is confirmed from the photo-assisted noise (at $\tau=0.77$).

In the PhD of Maelle Kapfer (<https://tel.archives-ouvertes.fr/tel-01978045>, 2019, Directed by C. Glattli) the same $F \sim 1/3$ was observed at $\nu=2/3$ on the $\tau=1/2$ plateau from the cross-correlation signal (independent of T, from 30mK to 100mK, whereas autocorrelations are increasing as T is reduced, see Fig. 7.10 page 116).

Here we observe $F \sim 1/3$ (for T=16, 31 and 40mK, focusing on cross-correlations at the lower T).

A possible explanation for the different observation in [49] at low temperature is that additional heat-induced noise is included in the measured autocorrelation signal (as pointed in e.g. [61] and in particular in Fig. 4b,d,f of this reference, see also the 'Filling factor 2/3' subsection of Methods and in particular Fig. 6c,f). In practice, this effect is reduced when the temperature is increased, which is related to the expected concomitant reduction of the thermal equilibration length between counter propagating modes. In the PhD of M. Kapfer, one can find a comparison of auto- and cross-correlations showing that the former is larger and increases as T decreases, consistent with the above interpretation, whereas the latter is stable with T (see Fig. 7.10). Moreover, when a comparison with a photo-assisted noise extraction of e^* is possible (tested at $\nu=2/5$ in Kapfer et al) an agreement was found with the cross-correlation Fano factor (but not with the auto-correlations if those were found larger than the cross-correlations). Accordingly, we focused here on the cross correlation signal in the specific low temperature case (T=16mK) where a relative increase of the auto-correlations is also observed (see Methods for further discussion).

In response to the referee, we now point out experimental works in reference to the observed quasiparticle charge $e/3$.

[Referee #2]

Minor points:

1. I could not find the data point in Fig. 3, which corresponds to Fig. 2d.

[Reply]

The data in Fig. 2d (and 2a) are measured on QPC_W with $V_{tg}=50mV$. The corresponding symbols showing the fitted charge and scaling dimension is in the supplementary information Fig. S4 (middle right panel). Figure 3 displays measurements on QPC_E.

In response to the referee, we now indicate in the caption of Fig. 2 the QPC used to obtain the displayed data.

[Referee #2]

2. I think “Note that in configurations used for investigating Δ , the noise measurement contacts are electrically upstream a floating contact (used to measure $\langle IB, F \rangle$) and, consequently, there is no such hot spot” is rather misleading. If I understand correctly, e.g., in Fig. 1, the ground contact where I_B enters provides a hot spot. But the generated heat cannot propagate to the noise measurement contact since the floating contact in the middle absorbs the heat.

[Reply]

The referee understood correctly. We have reformulated this misleading sentence according to the referee’s suggestion.

[Referee #2]

3. While I found the paragraph “(i) non-chiral heating” in Methods section very interesting, the present texts are not transparent.

[Reply]

The text of this section has been mostly rewritten for a much improved clarity in response to the referee’s request.

Referee #3

A. The authors report on measurements of low bias noise observed in quantum point contacts operated in the fractional quantum Hall regime. Their qpcs are fabricated on a two-dimensional electron gas residing in an AlGaAs/GaAs heterostructure. They extract the scaling dimension Δ for $\nu=1/3$, $\nu=2/5$, and $\nu=2/3$ by examining the crossover from thermal noise to shot noise via the change in the Fano factor.

B. The measurement of noise generated by partitioning edge states at qpcs in the fractional quantum Hall regime is not new - there are many reports in the literature. However the authors do appear to focus their analysis on new and interesting regime associated with the transition from thermal noise to shot noise at modest bias with the aim to extract the scaling dimension at each fraction. This aspect is certainly original and significant. Furthermore, the determination of the scaling dimension has taken on new importance in the study of anyons in the fractional quantum Hall regime. By this measure, the research is timely, original, and potentially highly impactful.

C. The data is of high quality and generated by established experts in the field. I do not doubt the veracity of their experimental results. I do request that the authors add some details for clarification and simplicity of understanding. For example, the authors should clearly state the definition of τ in equation 1 and equation 2. This is known to experts, but it should nevertheless be stated explicitly once in the main text.

[Reply]

In response to the referee, we have relocated the definition of tau to just underneath Eq. 1, where the reader may most naturally look for it. It is also provided within Fig. 1 (in the figure itself). Finally,

we added a paragraph in the 'Measurement' subsection of 'Methods' to provide explicit expression for the composite channel case of $\nu=2/5$.

[Referee #3]

Secondly, the authors should mention the error bars on their experimental determination of noise as a function of bias. It may be that the error bars are smaller than the points used to display the data, but it is never stated explicitly. Just showing data points without an error estimate implies perfect precision.

[Reply]

Error bars in the noise data (Fig. 2a,b,c) are indeed smaller than the symbols (much smaller at $\nu=1/3$, comparable for the most challenging inner channel of $\nu=2/5$).

In response to the referee, we now explicitly provide the experimental resolution (the standard error) on the displayed noise data within the figure caption.

[Referee #3]

Since precise knowledge of electron temperature in the device is also crucial, it would also be nice explicitly state the electron temperature for the data in Figure 2 rather than saying $T \approx 30\text{mK}$ in passing in the text. If the temperature is fixed at $T=31\text{mK}$, please include this information in the caption or directly in the figure.

[Reply]

We have added the precise, independently obtained temperatures (also used in the fits) in the caption of Fig. 2.

[Referee #3]

D. The use of statistics seems reasonable. Some of the uncertainties associated with the analysis may require elaboration. I discuss further in my subsequent comments.

E. The data and analysis is intriguing and suggestive that the authors have a reasonable method to measure the scaling dimension Δ in the fractional states. It also raises some questions. The qpc transmission vs. gate voltage at $1/3$ is full of resonances, forcing the authors to exclude certain points from the determination of the scaling dimension. This may be reasonable in this specific case, but it does uncertainty to the methodology. For example, in Fig. 3a the extracted charge is systematically low as the transmission is varied. Can the authors associate this observation with any systematic effect?

[Reply]

Most often, in the different labs investigating QPCs in the fractional quantum Hall regime, the transmission vs gate voltage at zero dc bias and low temperatures shows many resonances at $\nu=1/3$ (more than at $\nu=2/5$ and $2/3$).

From this observation, we know that some tunings are susceptible to display a quantum dot behavior (asymmetry in dc bias voltage, a possibly very different amount of noise, a strong energy dependence) rather than that of a QPC. Our methodology for automatically excluding these tunings is designed to limit any human bias, by considering the quantitative accuracy of the best possible fit of the data with Eq. 2 (leaving the values of e^* and Δ completely free). The referee possibly wonders if this strategy is sufficient. It is here validated by the remarkable robustness of our findings.

The scattering of the fitted data points remains relatively small, weakly dependent on the QPC used, on τ , on the presence of a surrounding gate and on the voltage applied to it, on temperature, or on the presence of a copropagating inner edge channel (at $\nu=2/5$).

In summary, we did not notice clear and systematic relations between deviations on e^* or Δ and some other characteristics of the tuning point, but rather a remarkable robustness on the considered data (for which a good fit with Eq. 2 is possible).

[Referee #3]

The scaling dimension is also evaluated over a large range of τ while strictly speaking equation 2 is only valid with $\tau \ll 1$. Can the authors comment on why extraction of Δ over such a broad range is justified?

[Reply]

The perturbative prediction is complemented by a factor $(1-\tau)$ accounting for correlations corrections beyond $\tau \ll 1$. This approach is successfully used for extracting the charge from shot noise measurements over a broader range of τ . The similar robustness in τ of the extracted Δ suggests similarly that the same correction applies for the thermal to shot noise crossover. However, like for shot noise, there is no direct derivation.

In response to the referee, we complemented the sentence below Eq. 2 by recalling that $1-\tau$ is a ad hoc factor added to extend the analysis beyond the tunneling regime where it is rigorously derived.

[Referee #3]

While the extracted Δ at $1/3$ filling factor is impressively close to the theoretical value of $1/6$, much more variation is observed at $\nu=2/5$ and $\nu=2/3$. At these fractions the extractions of the scaling dimension appear less robust. It would be helpful if the authors could comment on this observation. Notably, both $2/3$ and $2/5$ support multiple edge modes. Do interactions between the modes potentially lead to corrections to the scaling dimension?

[Reply]

We do not believe that the larger scatter of the data points for Δ on the inner channel of $\nu=2/5$ indicates less robustness. This scatter rather corresponds to a resolution that is not as high as for $\nu=1/3$, $\nu=2/3$ or the outer channel of $\nu=2/5$. Indeed, this inner channel only carries $1/6$ of the injected current, resulting in a much weaker noise signal to be compared with a similar resolution ($\sim 10^{-31} \text{ A}^2/\text{Hz}$, now provided in the caption of Fig. 2). This translates into a larger scatter of the fitted values of the scaling dimension (more than on the fitted values of the charge, which is determined by the stronger shot noise signal at higher bias).

From a theoretical standpoint, the quasiparticles scaling dimension is predicted to be robust to interactions between chiral channels co-propagating in the same direction along the same edge (see eg Appendix in [48]), such as for $\nu=2/5$. The negligible impact of inter-channel coupling at $\nu=2/5$ is here confirmed, at a higher experimental accuracy, by the data on the outer channel of $\nu=2/5$. Indeed, those match the same pristine quantum Hall channel prediction for Δ as the single $\nu=1/3$ channel.

The values of Δ extracted at $\nu=2/3$ data appear rather robust to us, including up to rather high τ where deviations are expected to eventually develop (see also Supplementary Fig. S5).

Theoretically, this state is however much more challenging, with counter-propagating channels and

several possible descriptions of the edge structure. In this non-fully chiral case, the coupling between modes has a strong impact on the scaling dimension. In practice, we find that our noise data are reasonably consistent with the wide spread picture by Kane, Fisher and Polchinski [50] of a single charge mode going downstream and a single neutral mode going upstream, where tunneling $e/3$ quasiparticles are predicted to have a scaling dimension $\Delta=1/3$.

In response to the referee, we now indicate the experimental uncertainty on the noise in Fig. 2, and we point out the particular complexity of the $\nu=2/3$ state referring to Methods where the discussion was improved.

[Referee #3]

Figure 4 is a nice summary of their findings over multiple qpcs and under different settings.

F. The claim of the paper is that the scaling dimension is determined unambiguously for several fractional states. While the data and analysis support this conclusion for the $1/3$ state, the data for $2/5$ and $2/3$ suggest the complications due to disorder and nonuniversal physics still play a role in these measurements. The manuscript would be strengthened if the authors discuss what is necessary to remove remaining non-idealities in the measurements and/or samples. The manuscript is valuable for its discussion of a new approach to determination of the scaling dimension. It will be even stronger if the authors can suggest steps necessary to make the technique even more reliable.

[Reply]

In response to the referee, we have added a specific discussion regarding the practical importance of intercalating HfO₂ underneath the metallic gate. We also emphasize the specific design of our QPCs, intended to reduce artifacts from a coupling between counter-propagating channels across the metal gates (made for this purpose about three times the depth of the 2DEG, and with a specific tip opening chosen as a compromise between reduced coupling and localized contact). Finally, our work strongly advocates the importance of unbiased data analysis and of repeating measurements in multiple configurations, especially in the fractional quantum Hall context where non-ideal behaviors are likely to emerge from many resonances.

[Referee #3]

G. The references are appropriate.

H. The paper is well written and discussion of data collection and analysis is clear. This is a valuable contribution to the field. My main suggestion is more discussion of why the analysis may deviate from theoretical expectations in specific cases.

[Reply]

In response to the referee we have added relevant information regarding our experimental resolution, which could explain the scatter of data points for the inner channel at $\nu=2/5$.

We also expanded our discussion of the theoretically more complex case of $\nu=2/3$, where other kinds of artifacts may emerge from the counter-propagating heat flow, and where the theoretical predictions for Δ is yet not unambiguously established (with multiple edge scenario presently debated).

Reviewer Reports on the First Revision:

Referees' comments:

Referee #1 (Remarks to the Author):

I understand that the scaling dimension is important for quantum Hall circuits. However, its importance and interest are difficult for general public to understand. Readers are interested in fractional charges because they usually think that charges are elementary charges. Also, readers are surprised at anyon statistics because boson and fermion statistics are well known to them. On the other hand, readers do not intuitively know what the scaling dimension is and why $\nu/2$ is trivial at all, when reading the modified manuscript. Therefore, I cannot judge whether this paper is likely to interest readers outside its own immediate field. I do not believe that this manuscript has "an exceptionally wide impact, both among scientists and, frequently, among the general public."

The discussion of the Majorana mode in nanowires was also mentioned in the authors' response. If this is the case, it should be mentioned in the manuscript along with some references.

Referee #2 (Remarks to the Author):

In the first round, I did not recommend the paper in Nature, based on (1) non-universality of scaling dimension (in a general perspective) by contrast with the charge and statistics of anyons. Furthermore, (2) their interpretation of the noise generation on the conductance plateau in the $\nu=2/3$ state does not match with the current theory; As the authors claim that the on-plateau noise (and hence the scaling dimension) is also attributed to the tunneling of anyons (the main theme of the paper), similar to the $\nu=1/3$ and $\nu=2/5$ states, it is important to figure out the underlying mechanism of the noise. However, I recommended the paper in Nature physics as I found it novel that the observation of scaling dimension of anyons may be used for time-resolved experiments including the recent experimental activities for measuring the braiding statistics of anyons.

In the revised version, I am not convinced by the author's reply to the points (1) and (2) as I will discuss on more details below. I recommend the paper in Nature physics if the authors provide suitable changes concerning the point (2).

Point (1): I disagree with the author's point that "We agree that, in general, the quasiparticles scaling dimension, and therefore the quasiparticles themselves, could be impacted by e.g. interaction between counter propagating channels. Note however that this would also be the case of the charge and statistics of 1D quasiparticles along such interacting edges (as illustrated with e.g. the charge fractionalization along nearby, counter-propagating integer quantum Hall edges observed in Kamata et al., Nature Nanotechnology 9, 177 (2014))"

The question would be what charge or statistics is a topological quantity in fractional quantum Hall

states (FQHS). Concerning the charge quantity, one can locally, in principle, add any amount of charge as you wish. An example is interaction-induced charges as illustrated by a reference Kamata et al., Nature Nanotechnology 9, 177 (2014) that authors quote. This quantity is not topological at all; it is just a property of Luttinger liquids. On the other hand, FQHS goes beyond this Luttinger liquid physics, possessing non-trivial topology. The genuine topological (charge) quantity of FQHS is the charge of elementary quasiparticles, which is identical with the basic unit of transferred quasiparticles across a quantum point contact through a quantum Hall bulk; It was experimentally shown in the seminal works, Refs. [3-4] (note that in the reference the author quoted, there is no charge tunneling at all). This quantity is highly universal, independent of the interaction strength on edges.

The obtained scaling dimensions by the authors are largely consistent with the value in the absence of interaction between the modes across a quantum point contact. But this is a result of fine-tuning of experiments and thus cannot be generic. It is not surprising that one measures different values of the scaling dimension from those obtained by the authors. I foresee such a non-universality also complicates the identification of tunneling quasiparticles by measuring the scaling dimension.

Point (2): I disagree with the authors that the noise generation on the conductance plateau $\nu=1/2$ in the $\nu=2/3$ state is attributed from a stochastic process of anyon tunneling. Shot noise measurements on a conductance plateau could naively be expected to produce no noise since no partitioning of the edge channels is expected (which is indeed the typical case in the integer quantum Hall regime). Thus, the measurements of finite noise on the plateau call for an explanation. Ref. [62] provides an explanation. Note that the mechanism proposed in Ref. [62] is based on the Kane-Fisher-Polchinski edge model without any edge reconstruction; one can identify $\nu_+=1$ and $\nu_-=-1/3$ in Fig. 2 of Ref. [62]. When a fractional quantum Hall liquid with different filling from the bulk filling factor is stabilized in a quantum point contact, the edge channels inside the quantum point contact (QPC) equilibrate with each other, forming a conductance plateau. As a result of equilibration, the Joule heating takes place in the QPC, and thus renders the local temperature near the QPC to be enhanced by $\Delta T \sim V$, where V is the applied bias voltage. On that plateau, the noise is generated due to the increased temperature, which is proportional to V . In particular, there are two distinct types of noise spots, see Fig. 4 in Ref. [62]; E and F near the contacts, C and D inside the QPC. Importantly, C and D contribute to both the cross-correlation and the auto-correlation since the current in the region C and D split to two currents, each of which arrives different drains D1 and D2. However, E and F only contribute to the auto correlation since the current partitioned in those regions arrives at the same drain, either D1 or D2. Note that this noise mechanism has nothing to do with anyon tunneling.

The authors claim that the above noise mechanism is inconsistent with their three observations: (i) the noise slope e^* weakly dependent on ambient temperatures of the system, (ii) the same value of auto-correlation and cross-correlation except for 16 mK, and (iii) weaker dependence of e^* on the transmission τ . But, the observations (i), (ii), and (iii) can be understood within the theory as follows:

(i) Thermal equilibration length should strongly depend on the applied voltage V , and only weakly on the ambient temperatures T_0 . As seen in the above mechanism, the application of V heats the quantum point contact region to an effective temperature $\Delta T \sim V$ and thus the characteristic

equilibration length l_{eq} scales with V in the regime of $V \gg T_0$, where the noise slope or equivalently e^* is extracted. The effect of the ambient temperature is weaker and it is only subdominant as $\Delta T \sim \sqrt{V^2 + T_0^2}$ in the regime of $V \gg T_0$.

(ii) Based on the above theory, the auto-correlation and the cross-correlation are indeed identical at higher temperatures. While all the noise spots C, D, E, F contribute to the auto correlation, the cross-correlation only comes from the noise spots C and D. Since C and D are located inside the QPC and they are geometrically close to the hot spots, the noise contribution from C and D is very weakly dependent on the equilibration length l_{eq} and thus ambient temperatures T_0 ; all generated heat activate the noise spots. However, E and F noise spots (near a contact) can only activate when the generated heat near the QPC arrives at the noise spots E and F; i.e., l_{eq} should be smaller than the distance between the QPC and the contact for providing a contribution to the auto correlation noise. Thus, the auto correlation can depend on T_0 ; at higher temperatures, only the noise spots C and D activate and contribute to both auto-correlation and cross-correlation. But at sufficiently lower temperatures, the noise spots E and F start to contribute to the auto-correlation, making the deviation between the auto-correlation and cross-correlation. Therefore, the cross-correlation is independent of T_0 , whereas the auto correlation increases with lowering ambient temperatures. This observation is highly supported by the noise measurement in the N-C configuration for Fig. 6b where only at the lowest accessible temperature $T_0=16\text{mK}$, the noticeable noise has been measured since the generated heat at the QPC can reach the contacts only at this temperature.

Finally, the observation (iii) is also within the theory. At exactly half-transmission ($\tau=1/2$), the noise should have a maximum value since the heat is maximally generated at the transmission. Deviating from the conductance plateau, the heating is reduced monotonously and hence the noise and e^* also change monotonously. As a side remark, I note that the noise at low transmission regime (small τ) is unambiguously generated by the stochastic tunneling of anyons as the authors proposed. In this transmission window, the heating at the QPC is not expected since the transmission is far away from the conductance plateau. The authors observed that near $\tau \sim 0.1$, the Fano factor and the scaling dimension have a lower value than the value in the vicinity of half-transmission, see Fig. 3c. This may indicate two independent mechanisms depending on the value of transmission.

Overall, I offer the authors to provide a noise mechanism where anyon tunneling can be compatible with the formation of the conductance plateau and the generation of noise, if the authors do not agree with the above mechanism for the noise near the conductance plateau. Presenting a mechanism at least, on a qualitative level, is now very important since the authors claim that the physics of $\tau=1/2$ plateau in the $\nu=2/3$ state is attributed to the anyon tunneling similar to the $\nu=1/3$ and $\nu=2/5$ case in contrast with the current theory.

Referee #3 (Remarks to the Author):

This is the 2nd report on this manuscript so I will focus on the authors' response to the questions and concerns from the first round of review.

I will first focus on the authors' responses to my questions and then to the comments of the other referees.

The authors have attempted to address my straightforward requests concerning definitions, error bars, clarification of measurement temperature, etc. These aspects have been addressed to my satisfaction. I thank the authors for this.

Their responses to some of the more fundamental questions were less satisfying and the authors are a bit more confident in the "robustness" and universality of their findings than I would be at this point. There are still several aspects of the transport properties of the qpcs reported here that do not agree with standard theoretical predictions. To these questions, the authors' responses indicate uncertainty remains that will not be answered in this work. When pressed on why their charge determination at $\nu=1/3$ appears systematically low, the response was that this feature has been reported in the literature previously. It may have been, but this is not an answer to my question. As to why the fitting may be applied over such a broad range of transmission, the authors claim the ad hoc addition of an $(1-\tau)$ factor to a phenomenological expression allows for this. Ok, but why does it work?

To be clear, there isn't anything really new in the measurements or sample design itself, but rather, the novelty is in the application of analysis of the Fano factor in the low bias regime to extract the scaling dimension rather than focusing solely on the high bias regime to extract the charge. They seem to have applied this approach first and have done the best they can to be systematic in their analysis. This is the strength of the paper. Questions will of course remain, and require more experiments and theoretical analysis. Nevertheless I feel the work is valuable to the community and should be published. I am comfortable with the scientific claims.

The other referees do raise the question if this manuscript is suitable for Nature or a more specialized journal. I can appreciate their points, but I am not too worried about this. The paper may be published in Nature or Nature Physics (the original braiding experiment at $1/3$ was published in Nature Physics after all). I think the work is solid and important enough to be published in Nature. The strength of the findings is what matters most to me and I believe this work crosses this threshold.

Author Rebuttals to First Revision:

We would like to thank the referees' for their suggestions, comments and questions.

Referee #1

I understand that the scaling dimension is important for quantum Hall circuits. However, its importance and interest are difficult for general public to understand. Readers are interested in fractional charges because they usually think that charges are elementary charges. Also, readers are surprised at anyon statistics because boson and fermion statistics are well known to them. On the other hand, readers do not intuitively know what the scaling dimension is and why $\nu/2$ is trivial at all, when reading the modified manuscript. Therefore, I cannot judge whether this paper is likely to interest readers outside its own immediate field. I do not believe that this manuscript has "an exceptionally wide impact, both among scientists and, frequently, among the general public."

The discussion of the Majorana mode in nanowires was also mentioned in the authors' response. If this is the case, it should be mentioned in the manuscript along with some references.

[Reply]

We have added a dedicated remark and the new reference [15] in the first paragraph of the manuscript (second sentence).

Referee #2

In the first round, I did not recommend the paper in Nature, based on (1) non-universality of scaling dimension (in a general perspective) by contrast with the charge and statistics of anyons. Furthermore, (2) their interpretation of the noise generation on the conductance plateau in the $\nu=2/3$ state does not match with the current theory; As the authors claim that the on-plateau noise (and hence the scaling dimension) is also attributed to the tunneling of anyons (the main theme of the paper), similar to the $\nu=1/3$ and $\nu=2/5$ states, it is important to figure out the underlying mechanism of the noise. However, I recommended the paper in Nature physics as I found it novel that the observation of scaling dimension of anyons may be used for time-resolved experiments including the recent experimental activities for measuring the braiding statistics of anyons.

In the revised version, I am not convinced by the author's reply to the points (1) and (2) as I will discuss on more details below. I recommend the paper in Nature physics if the authors provide suitable changes concerning the point (2).

Point (1): I disagree with the author's point that "We agree that, in general, the quasiparticles scaling dimension, and therefore the quasiparticles themselves, could be impacted by e.g. interaction between counter propagating channels. Note however that this would also be the case of the charge and statistics of 1D quasiparticles along such interacting edges (as illustrated with e.g. the charge fractionalization along nearby, counter-propagating integer quantum Hall edges observed in Kamata et al., Nature Nanotechnology 9, 177 (2014))"

The question would be what charge or statistics is a topological quantity in fractional quantum Hall states (FQHS). Concerning the charge quantity, one can locally, in principle, add any amount of charge as you wish. An example is interaction-induced charges as illustrated by a reference Kamata et al., Nature Nanotechnology 9, 177 (2014) that authors quote. This quantity is not topological at all; it is just a property of Luttinger liquids. On the other hand, FQHS goes beyond this Luttinger liquid physics, possessing non-trivial topology. The genuine topological (charge) quantity of FQHS is the charge of elementary quasiparticles, which is identical with the basic unit of transferred

quasiparticles across a quantum point contact through a quantum Hall bulk; It was experimentally shown in the seminal works, Refs. [3-4] (note that in the reference the author quoted, there is no charge tunneling at all). This quantity is highly universal, independent of the interaction strength on edges.

The obtained scaling dimensions by the authors are largely consistent with the value in the absence of interaction between the modes across a quantum point contact. But this is a result of fine-tuning of experiments and thus cannot be generic. It is not surprising that one measures different values of the scaling dimension from those obtained by the authors. I foresee such a non-universality also complicates the identification of tunneling quasiparticles by measuring the scaling dimension.

[Reply]

We agree with the referee statement that the quasiparticle excitations in the gapped fractional quantum Hall bulk are topologically protected, in contrast to edge quasiparticles. Despite the many possible complications, we find here, for the QPC geometries, the sample preparation and the observables chosen to limit these complications, a robust agreement with genuine theoretical predictions. By robust, we mean that consistent observations were made for a broad range of gate voltage tunings, different magnetic fields, different local electronic densities, different temperatures, different physical QPC and in the presence or not of a large surrounding screening gate and of a co-propagating channel. We would not qualify this as an observation limited to a “fine-tuning of experiments”, but we would rather argue that this constitutes a demonstration of a practical path to observe genuine properties of fractional quantum Hall quasiparticles.

Note that in the sentence of our answer quoted by the referee, our point was that the absence of protection “along the interacting edges” is not limited to the scaling dimension but extends to the charge and statistics of edge quasiparticles. The Kamata reference provides an illustration of interacting edges where an incoming quasiparticle of charge e would be split into two quasiparticles of different charges. Charge fractionalization along the edge is also known to take place along co-propagating interacting edges, eg at filling factor 2. Recently, we experimentally observed signatures of fractional statistics of such fractionalized charges along integer quantum Hall edge channels (arXiv:2401.06069). In the fractional quantum Hall regime, the fractional statistics signatures in ‘colliders’ and in interferometers [5,6] also involve edge quasiparticles (circling around bulk quasiparticles in the interferometer implementation), which are not topologically protected.

[Referee #2]

Point (2): I disagree with the authors that the noise generation on the conductance plateau $\nu=1/2$ in the $\nu=2/3$ state is attributed from a stochastic process of anyon tunneling. Shot noise measurements on a conductance plateau could naively be expected to produce no noise since no partitioning of the edge channels is expected (which is indeed the typical case in the integer quantum Hall regime). Thus, the measurements of finite noise on the plateau call for an explanation. Ref. [62] provides an explanation. Note that the mechanism proposed in Ref. [62] is based on the Kane-Fisher-Polchinski edge model without any edge reconstruction; one can identify $\nu_+=1$ and $\nu_-=-1/3$ in Fig. 2 of Ref. [62]. When a fractional quantum Hall liquid with different filling from the bulk filling factor is stabilized in a quantum point contact, the edge channels inside the quantum point contact (QPC) equilibrate with each other, forming a conductance plateau. As a result of equilibration, the Joule heating takes place in the QPC, and thus renders the local temperature near the QPC to be enhanced by $\Delta T \sim V$, where V is the applied bias voltage. On that plateau, the noise is generated due to the increased temperature, which is proportional to V . In particular, there are two distinct types of noise spots, see Fig. 4 in Ref. [62]; E and F near the contacts, C and D inside the QPC. Importantly, C and D

contribute to both the cross-correlation and the auto-correlation since the current in the region C and D split to two currents, each of which arrives different drains D1 and D2. However, E and F only contribute to the auto correlation since the current partitioned in those regions arrives at the same drain, either D1 or D2. Note that this noise mechanism has nothing to do with anyon tunneling.

The authors claim that the above noise mechanism is inconsistent with their three observations: (i) the noise slope e^* weakly dependent on ambient temperatures of the system, (ii) the same value of auto-correlation and cross-correlation except for 16 mK, and (iii) weaker dependence of e^* on the transmission τ . But, the observations (i), (ii), and (iii) can be understood within the theory as follows:

(i) Thermal equilibration length should strongly depend on the applied voltage V , and only weakly on the ambient temperatures T_0 . As seen in the above mechanism, the application of V heats the quantum point contact region to an effective temperature $\Delta T \sim V$ and thus the characteristic equilibration length l_{eq} scales with V in the regime of $V \gg T_0$, where the noise slope or equivalently e^* is extracted. The effect of the ambient temperature is weaker and it is only subdominant as $\Delta T \sim \sqrt{V^2 + T_0^2}$ in the regime of $V \gg T_0$.

(ii) Based on the above theory, the auto-correlation and the cross-correlation are indeed identical at higher temperatures. While all the noise spots C, D, E, F contribute to the auto correlation, the cross-correlation only comes from the noise spots C and D. Since C and D are located inside the QPC and they are geometrically close to the hot spots, the noise contribution from C and D is very weakly dependent on the equilibration length l_{eq} and thus ambient temperatures T_0 ; all generated heat activate the noise spots. However, E and F noise spots (near a contact) can only activate when the generated heat near the QPC arrives at the noise spots E and F; i.e., l_{eq} should be smaller than the distance between the QPC and the contact for providing a contribution to the auto correlation noise. Thus, the auto correlation can depend on T_0 ; at higher temperatures, only the noise spots C and D activate and contribute to both auto-correlation and cross-correlation. But at sufficiently lower temperatures, the noise spots E and F start to contribute to the auto-correlation, making the deviation between the auto-correlation and cross-correlation. Therefore, the cross-correlation is independent of T_0 , whereas the auto correlation increases with lowering ambient temperatures. This observation is highly supported by the noise measurement in the N-C configuration for Fig. 6b where only at the lowest accessible temperature $T_0 = 16\text{mK}$, the noticeable noise has been measured since the generated heat at the QPC can reach the contacts only at this temperature.

Finally, the observation (iii) is also within the theory. At exactly half-transmission ($\tau = 1/2$), the noise should have a maximum value since the heat is maximally generated at the transmission. Deviating from the conductance plateau, the heating is reduced monotonously and hence the noise and e^* also change monotonously. As a side remark, I note that the noise at low transmission regime (small τ) is unambiguously generated by the stochastic tunneling of anyons as the authors proposed. In this transmission window, the heating at the QPC is not expected since the transmission is far away from the conductance plateau. The authors observed that near $\tau \sim 0.1$, the Fano factor and the scaling dimension have a lower value than the value in the vicinity of half-transmission, see Fig. 3c. This may indicate two independent mechanisms depending on the value of transmission.

Overall, I offer the authors to provide a noise mechanism where anyon tunneling can be compatible with the formation of the conductance plateau and the generation of noise, if the authors do not agree with the above mechanism for the noise near the conductance plateau. Presenting a mechanism at least, on a qualitative level, is now very important since the authors claim that the physics of $\tau = 1/2$ plateau in the $\nu = 2/3$ state is attributed to the anyon tunneling similar to the $\nu = 1/3$ and $\nu = 2/5$ case in contrast with the current theory.

[Reply]

We thank the referee for his in-depth argumentation regarding the interpretation of the noise at $\tau^{1/2}$ for $\nu=2/3$, which helped us to improve our discussion.

Before answering in details, we would like to stress two points:

- This discussion regarding the noise mechanism at $\tau^{1/2}$ does not impact our extraction of the scaling dimension at small τ at $\nu=2/3$, as specifically pointed out above by the referee.
- We do not exclude a heating interpretation of the noise at $\tau^{1/2}$ for $\nu=2/3$.

In response to the referee, we have in particular modified the main text (section ‘Robustness of observations’) to warn the reader that “the noise interpretation is not as straightforward, especially when τ is not small” at $\nu=2/3$. We have also modified the discussion in Methods to provide a more balanced description of the heating and partition interpretations of the noise. This was achieved by putting forward one observation questioning the partition interpretation of the noise at $\tau=1/2$ (the presence of a small ‘plateau’) and one observation questioning the heating interpretation of the noise (the observed amount of noise at $\tau^{1/2}$).

We answer below on the different parts of the referee’s argumentation.

- First paragraph (summary of [63], previous reference [62]):

We thank the referee for the clear summary. We have removed from Methods a too restrictive description of the edge channel structure, as in the full equilibration model (specifically put forward by the referee) all effects of edge reconstruction are eliminated.

- (i) Effect of temperature on thermal equilibration length:

We accept the referee’s point that the effect of base temperature on equilibration length is not straightforward at large bias voltage, and modified the discussion in Methods accordingly.

- (ii) Auto vs cross-correlations:

We previously chose to put forward the comparison between auto- and cross-correlations as, according to the summary Table I in arXiv:2307.05173, this was a rather discriminatory observation (see e.g. the substantial constant term +0.09 with the same sign for both auto and cross-correlations and independent of the ratio L_{eq}/L_{arm} [L_{eq} : thermal equilibration length, L_{arm} : edge distance QPC-source ohmic contact] appearing for both full or mixed equilibrated regimes, see also Eqs. B11 and B12 in [63]).

Yet, the referee argues that the equality between auto- and cross-correlations may in fact be more generally expected for long enough arms. We accept this comment and removed the weakened argument from the discussion in Methods.

- (iii) Correlations between noise signal and τ :

The referee suggests in his answer that the effective Fano factor introduced by heating may evolve weakly with τ in the vicinity of the $\tau \sim 0.5$ plateau. Although, to the best of our knowledge, this statement goes beyond the situations addressed in the literature, we do not wish to rule out this possibility. As also pointed out by the referee, it remains that a heating noise mechanism at $\tau^{1/2}$

would involve a transition from the different tunneling noise mechanism at low transmission. These different mechanisms should generally result in different noise signals, eg different high bias slopes (e^* fit parameters) and different crossover shapes (Δ fit parameters).

In his remark, the referee states that e^* and Δ are smaller at low transmissions, which would fit with a different underlying mechanism. This statement may be based on the specific transmission dependence shown in Fig. 3c of the main text. We think it does not match the more general picture (see also the complementary data of Supplementary Fig. S5; at the lowest $\tau \sim 0.1$ the scatter of the data is larger): We rather observe similar e^* and Δ at the experimental accuracy over most of the explored range of τ , including $\tau \sim 0.5$.

Such weak changes from small τ to $\tau \sim 0.5$ would be most particularly surprising assuming that the full equilibrated model of heating noise put forward by the referee applies at $\tau = 0.5$, since a markedly super-poissonian noise is expected in this case: the high bias voltage prediction in this model is $F \sim (L_{arm}/L_{eq})^{0.5}$ [see eq. B8 in [63] with here, $L_{arm} \sim 150 \mu\text{m}$ and an equilibration length $l_{eq} \ll L_{qpc} \sim 1 \mu\text{m}$ under the full equilibration hypothesis]. Although it may in principle be reduced by bulk heat leakage, the latter was recently found to be very small at $\nu = 2/3$ [Melcer et al., Nature Physics 19, 327 (2023)].

Therefore, the observed rather weak variations of e^* and Δ over the broad explored range of τ questions the heating interpretation of the noise at $\tau \sim 0.5$, but it does not rule it out. We have modified the discussion in Methods accordingly.

- Alternative mechanisms (final paragraph):

The $\tau \sim 1/2$ 'plateaus' observed in our QPCs are rather narrow in gate voltage and not very well defined (see inset in Fig. 3c for QPC_E and the new supplementary figure S7 for QPC_SE). This leaves open the possibility for a simple, accidental explanation of the 'plateau', such as the specific way the barrier potential deforms with V_{gate} or from the effect of impurities nearby the QPC. Yet, such an accidental effect is generally unlikely and we are not aware of a more generic (non-accidental) explanation compatible with a partition interpretation of the noise signal at $\tau \sim 1/2$ and $\nu = 2/3$. Therefore, the presence of $\tau \sim 0.5$ 'plateaus' indeed questions the partition interpretation of the noise, but their limited flatness does not allow to rule out this interpretation.

We have revised the discussion in Methods to more clearly state this point.

Referee #3

This is the 2nd report on this manuscript so I will focus on the authors' response to the questions and concerns from the first round of review.

I will first focus on the authors' responses to my questions and then to the comments of the other referees.

The authors have attempted to address my straightforward requests concerning definitions, error bars, clarification of measurement temperature, etc. These aspects have been addressed to my satisfaction. I thank the authors for this.

Their responses to some of the more fundamental questions were less satisfying and the authors are a bit more confident in the "robustness" and universality of their findings than I would be at this

point. There are still several aspects of the transport properties of the qpcs reported here that do not agree with standard theoretical predictions. To these questions, the authors' responses indicate uncertainty remains that will not be answered in this work. When pressed on why their charge determination at $\nu=1/3$ appears systematically low, the response was that this feature has been reported in the literature previously. It may have been, but this is not an answer to my question. As to why the fitting may be applied over such a broad range of transmission, the authors claim the ad hoc addition of an $(1-\tau)$ factor to a phenomenological expression allows for this. Ok, but why does it work?

[Reply]

The referee will probably agree that our observations are robust in the sense that we obtained consistent findings for a broad range of gate voltage tunings, different magnetic fields, different local electronic densities, different temperatures, different physical QPC and in the presence or not of a large surrounding screening gate and of a co-propagating channel. Note that although not perfectly accurate, the agreement observed with genuine theoretical predictions remains rather remarkable, at least in our view, and extends to three different types of fractional quasiparticles.

Although we obtain consistent Δ and e^* values also beyond the tunneling limit ($\tau \ll 1$), where the ad-hoc $(1-\tau)$ factor is significant, we do agree that the scaling dimension and charge of the quasiparticles can be most straightforwardly extracted from the data at low τ , when this factor is negligible. This is explicitly specified several times in the manuscript (see page 4 below Eq. 2, page 5 in the left column at the level of Fig. 4, and now also page 5 near the bottom of the left column for the specific case of $\nu=2/3$).

[Referee #3]

To be clear, there isn't anything really new in the measurements or sample design itself, but rather, the novelty is in the application of analysis of the Fano factor in the low bias regime to extract the scaling dimension rather than focusing solely on the high bias regime to extract the charge. They seem to have applied this approach first and have done the best they can to be systematic in their analysis. This is the strength of the paper. Questions will of course remain, and require more experiments and theoretical analysis. Nevertheless I feel the work is valuable to the community and should be published. I am comfortable with the scientific claims.

The other referees do raise the question if this manuscript is suitable for Nature or a more specialized journal. I can appreciate their points, but I am not too worried about this. The paper may be published in Nature or Nature Physics (the original braiding experiment at $1/3$ was published in Nature Physics after all). I think the work is solid and important enough to be published in Nature. The strength of the findings is what matters most to me and I believe this work crosses this threshold.

List of changes

- Changes in the text are displayed in the marked manuscript (new text is in red, removed text is struck through).
- The reference [15] was added.

- In Fig. 6d,e,f of Methods, a numerical error in the scale of the horizontal axis (V) was corrected.
- In the Supplementary Information, the new Supplementary Fig. 7 was added, together with the last sentence of section II (Supplementary Data) pointing to this figure.

Reviewer Reports on the Second Revision:

Referees' comments:

Referee #2 (Remarks to the Author):

In the last round, I have raised two points: (1) the scaling dimension of anyons is not a topological property that, in principle, depends on microscopic details and (2) the nature of noise at half transmission $\tau \simeq 1/2$.

Concerning point (2), I agree with more fair and honest evaluation of the present manuscript that "Which picture more adequately describes the QPC at $\tau \simeq 1/2$ and $\nu = 2/3$ is not straightforward.". I also agree with the authors that understanding of the origin of noise at $\tau \simeq 1/2$ is not important for conveying the key message of the paper; At small τ , noise is unambiguously produced by the tunneling of anyons across a quantum Hall bulk region. I have raised this question as the previous version was written in a way which may mislead readers about the nature of noise at $\tau \simeq 1/2$. I still would like to point out that the main figures Fig. 2c and 2f employ the data at $\tau \sim 0.43$ (close to $\tau = 0.5$). I recommend those figures to replace with data at lower τ if possible. Otherwise, I am fine with this point.

Point (1) was the main basis upon which I only recommend the paper in Nature physics. In the reply letter, they emphasized the robustness of the measured scaling dimension which they claimed to be consistent with the theoretical predictions. But the predictions themselves are a result of fine-tuning of interactions at QPC, e.g., neglecting the interaction between counter-propagating modes across a QPC. I really appreciate the systematic check of robustness of their observation (this is one of the underlying grounds to warrant this paper published in a high-profile journal, e.g., Nature physics, see below). But as stated in the added note, different groups measured different values of the scaling dimension and the authors admitted it may be "related to differences in the geometry of the QPCs"; then I do not see how the authors can emphasize the robustness of their observation.

Furthermore, they stated in the reply that "our point was that the absence of protection "along the interacting edges" is not limited to the scaling dimension but extends to the charge and statistics of edge quasiparticles". But the "absent protection" of the charge and statistics along interacting edges has nothing to do with their topological aspect. The genuine topological excitations are anyons in the bulk. Their topological properties (charge and statistics) are protected by the bulk gap while the scaling dimension of anyons is non-universal depending on their coupling to any other degrees of freedom. The charge and statistics of anyons are allowed to be measured by edge probes only when anyonic excitations on edges tunnel across the quantum Hall bulk region. Apart from this situation, nothing is topologically protected on edges, but this fact does not imply that the charge and statistics of bulk quasiparticles are not topological. I do not think such non-universality is appealing to general readership and hence I do not recommend the paper in Nature.

Nevertheless, I found the paper interesting: (i) probably first paper to reveal a dynamical property of anyons, so-called the scaling dimension. The observation of this quantity allows to understand various time-dependent dynamical experiments. It also may provide a useful information for extracting the braiding statistics of anyons. (ii) The authors checked their observations systematically to show how their novel method (crossover of thermal noise to shot noise) works in many different experimental conditions. Based on (i) and (ii), I strongly recommend the paper published in Nature physics.